# Discrete Theta Angles, Symmetries and Anomalies

Po-Shen Hsin[1] and Ho Tat Lam[2]

[1] Walter Burke Institute for Theoretical Physics,

California Institute of Technology, Pasadena, CA 91125, USA

[2] Physics Department, Princeton University, Princeton, NJ 08540, USA

## Abstract

Gauge theories in various dimensions often admit discrete theta angles, that arise from gauging a global symmetry with an additional symmetry protected topological (SPT) phase. We discuss how the global symmetry and 't Hooft anomaly depends on the discrete theta angles by coupling the gauge theory to a topological quantum field theory (TQFT). We observe that gauging an Abelian subgroup symmetry, that participates in symmetry extension, with an additional SPT phase leads to a new theory with an emergent Abelian symmetry that also participates in a symmetry extension. The symmetry extension of the gauge theory is controlled by the discrete theta angle which comes from the SPT phase. We find that discrete theta angles can lead to two-group symmetry in $4d$ QCD with $SU(N), SU(N)/\mathbb{Z}_k$ or $SO(N)$ gauge groups as well as various $3d$ and $2d$ gauge theories.

Tuesday 8$^{\text{th}}$ September, 2020

# 1   Introduction

Gauge theories often admit topological terms that assign different weights to different bundles in the partition function

$$Z = \sum_v \alpha_v Z_v \ , \tag{1.1}$$

where $v$ denotes different topological sectors. Some sectors might be absent in the sum if $\alpha_v$ vanishes (see the examples in [1–3][1]). If $\alpha_v$ is nonzero, it can be a discrete phase in some

---

[1]Some models are also discussed in [4] and the references therein.

theories. We will refer to it as a discrete theta angle. Some examples were presented in [3]. In this note we discuss the general relation among families of gauge theories with different discrete theta angles. In particular, we will focus on their global symmetries and 't Hooft anomalies.[2]

Theories with different discrete theta angles often arise from gauging a global symmetry in a quantum field theory with different symmetry-protected topological (SPT) phases. Gauging the symmetry sums over different topological sectors labelled by the gauge field. Let us denote two SPT phases by $\mathcal{S}$ and $\mathcal{S}'$ with partition functions $\alpha_v, \alpha_v'$, and their resulting theories after gauging the symmetry by $\mathcal{T}$ and $\mathcal{T}'$. In such cases, the theories $\mathcal{T}$ and $\mathcal{T}'$ are related by coupling to a topological quantum field theory (TQFT). The TQFT is constructed by gauging the global symmetries in the SPT phase $(\mathcal{S}' - \mathcal{S})$ with the partition function $\alpha_v(\alpha_v')^*$ by summing over the topological sectors,

$$Z_{\text{TQFT}} = \sum_v \alpha_v(\alpha_v')^* \ . \tag{1.2}$$

When the symmetry being gauged is Abelian (for simplicity we will assume it to be discrete), the theories $\mathcal{T}$, $\mathcal{T}'$ as well as the TQFT has a dual non-anomalous Abelian symmetry $\mathcal{A}$. Gauging the symmetry in the theories $\mathcal{T}$ and $\mathcal{T}'$ restricts the sum over the topological sector to a single term and recovers the original theory. More generally, one can use the dual symmetry $\mathcal{A}$ to couple the theory $\mathcal{T}$ to the TQFT by gauging the diagonal symmetry

$$\mathcal{T}' \quad \longleftrightarrow \quad \frac{\mathcal{T} \times \text{TQFT}}{\mathcal{A}} \ . \tag{1.3}$$

The coupling identifies the gauge fields in the theory $\mathcal{T}$ and in the TQFT so the theory after gauging is equivalent to $\mathcal{T}'$ with a different discrete theta angles.

We can then determine the properties of the theory $\mathcal{T}'$ from the theory $\mathcal{T}$ and the TQFT. Theories with different discrete theta angles form a family of theories. The difference between theories within a family is captured universally by the TQFTs that relate them. In this note we study these universal aspects that depend on the TQFTs.[3] We discuss several examples including gauge theories with or without matter in $3d$ and $4d$.

In some examples, the symmetry that we gauge is a subgroup of a larger symmetry. If the larger symmetry is a non-trivial extension of the gauged subgroup and its quotient (in other words, not a direct product), we observe that the resulting gauge theories have different extensions of global symmetries and 't Hooft anomalies, that depend on the SPT phases *i.e.* the discrete theta angles for the gauged symmetry.

---

[2]We will also present examples where theories with different discrete theta angles differ in their non-invertible topological defects, see Section 5.4.

[3]For $4d$ theories a similar construction is discussed in [5] that studies different symmetry fractionalizations using the TQFT sector.

When the discrete theta angle vanishes, our results agree with the general discussion in [6], where a mixed anomaly is observed in the resulting gauge theory due to the symmetry extension in the original theory. On the other hand, for nonzero discrete theta angle we find such mixed anomaly can be absent.

When the symmetries involved in the extension are $q$-form symmetries with different degrees $q$ [7], the global symmetry describes a higher-group [8–10]. We stress that in order to produce the symmetry extension, the original symmetry does not need to have an anomaly (and adding an SPT phase also does not change the anomaly of the theory ). This is a generalization of the discussion in [11, 6, 9, 10], which describes a special case (gauging a symmetry without adding an SPT phase). It was shown that a mixed anomaly in the original symmetry produces a symmetry extension in the gauge theory, while here we find that the mixed anomaly is not necessary for the symmetry extension in the gauge theory. In particular, we show that theories with two-group symmetries can be constructed by gauging a subgroup symmetry that does not have a mixed anomaly with the remaining symmetry.

We use the method to study the global symmetry and its 't Hooft anomaly in various theories, including $3d$ gauge theory and $4d$ $SU(N)/\mathbb{Z}_k$ and $SO(N), Spin(N), O(N)$ gauge theories. $SU(N)/\mathbb{Z}_k$ gauge theory in $4d$ has a discrete theta angle $p$ with even $pk$ [12, 13]

$$\frac{2\pi p}{2k} \int \mathcal{P}(w_2^k), \quad p = 0, 1, \cdots 2k - 1 , \tag{1.4}$$

where $w_2^k$ is the obstruction to lifting the bundle to an $SU(N)$ bundle, and $\mathcal{P}$ is the Pontryagin square operation reviewed in Appendix B [14]. $SO(N)$ gauge theory in $4d$ has a $\mathbb{Z}_4$ discrete theta angle [12]

$$\frac{2\pi p}{4} \int \mathcal{P}(w_2^{(1)}), \quad p = 0, 1, 2, 3 , \tag{1.5}$$

where $w_2^{(1)}$ is the obstruction to lifting the gauge bundle to a $Spin(N)$ bundle. $O(N)$ gauge theory in $4d$ has the discrete theta angle

$$\frac{2\pi p}{4} \int \mathcal{P}(w_2^{(1)}) + \pi r \int (w_1)^2 w_2^{(1)}, \quad p = 0, 1, 2, 3, \quad r = 0, 1 , \tag{1.6}$$

where $w_1, w_2^{(1)}$ are the first and second Stiefel-Whitney classes of the $O(N)$ bundle. The TQFTs corresponding to these discrete theta angles are two-form and one-form gauge theories (see Section 2 and Appendix C for details). In particular, we find two-group symmetries or symmetry extension that depends on the discrete theta angles of the gauge theory. Some examples are

- $4d$ $SU(N)/\mathbb{Z}_k$ gauge theory with discrete theta angle $p$ and $N_f$ massless Dirac fermions

in the tensor representation with $r$ boxes in the Young tableaux that satisfies the relation $\gcd(N, r) = k$. The theory has $\mathbb{Z}_k$ magnetic one-form symmetry, and the flavor 0-form symmetry

$$G^{(0)} = \frac{\widetilde{G}^{(0)}}{\mathbb{Z}_{N/k}}, \quad \widetilde{G}^{(0)} = \frac{SU(N_f)_L \times SU(N_f)_R \times U(1) \times \mathbb{Z}_{2N_f I(\mathcal{R})}}{\mathbb{Z}_{N_f} \times \mathbb{Z}_{N_f} \times \mathbb{Z}_2} \ . \quad (1.7)$$

where the various quotients are explained in section 3.3. The one-form symmetry and the flavor symmetry combines into a two-group symmetry with Postnikov class[4]

$$\Theta = p\mathrm{Bock}(w_2^f) \ , \quad (1.9)$$

where $w_2^f$ is the obstruction associated with the $\mathbb{Z}_{N/k}$ quotient in the flavor symmetry background gauge field, and Bock is the Bockstein homomorphism for the short exact sequence $1 \to \mathbb{Z}_k \to \mathbb{Z}_N \to \mathbb{Z}_{N/k} \to 1$.

- $4d$ $SO(N)$ gauge theory with discrete theta angle $p$ and $N_f$ massless Weyl fermions in the vector representation, with even $N$ and $N_f$. The theory has a flavor symmetry

$$G^{(0)} = \frac{\widetilde{G}^{(0)}}{\mathbb{Z}_2}, \quad \widetilde{G}^{(0)} = \frac{SU(N_f) \times \mathbb{Z}_{2N_f}}{\mathbb{Z}_{N_f}} \ , \quad (1.10)$$

and $\mathbb{Z}_2$ charge conjugation symmetry that extends the $SO(N)$ gauge field to $O(N)$ gauge field. The theory also has a $\mathbb{Z}_2$ magnetic one-form symmetry. The symmetries combine into a two-group symmetry with Postnikov class

$$\Theta = p\left(\frac{N}{2}\mathrm{Bock}(w_2^f) + w_2^f B_1^{\mathcal{C}}\right) \ , \quad (1.11)$$

where $w_2^f$ is the obstruction associated with the $\mathbb{Z}_2$ quotient in the flavor symmetry, and $B_1^{\mathcal{C}}$ is the background gauge field of the charge conjugation symmetry. Bock denotes the Bockstein homomorphism for the short exact sequence $1 \to \mathbb{Z}_2 \to \mathbb{Z}_4 \to Z_2 \to 1$.

- $3d$ $\mathbb{Z}_N$ gauge theory with discrete theta angle given by Chern-Simons level $k$, obtained by gauging a $\mathbb{Z}_N$ normal subgroup 0-form symmetry in a system with $\widetilde{G}$ 0-form

---

[4]For one-form symmetry $G^{(1)}$ and 0-form symmetry $G^{(0)}$, the Postnikov class $\Theta \in H^3(G^{(0)}, G^{(1)})$ expresses how the zero-form and one-form symmetries combine in terms of their backgrounds $B_1, B_2$

$$\delta B_2 = B_1^* \Theta \ , \quad (1.8)$$

where $\delta$ is the differential (the coboundary operator for $C^*(M, G^{(1)})$ on spacetime $M$) and $B_1^* \Theta$ is the pullback of $\Theta$.

symmetry. $\widetilde{G}$ is the group extension $1 \to \mathbb{Z}_N \to \widetilde{G} \to G \to 1$ described by $\eta_2 \in H^2(G, \mathbb{Z}_N)$. We assume the $\mathbb{Z}_N$ subgroup symmetry is non-anomalous and there is no mixed anomaly between $\mathbb{Z}_N$ and $\widetilde{G}$. The new theory has two-group symmetry that combines the emergent $\mathbb{Z}_N$ dual one-form symmetry generated by the Wilson line and $G$ 0-form symmetry, with the Postnikov class

$$\Theta = k\mathrm{Bock}(\eta_2) \ , \tag{1.12}$$

where Bock is the Bockstein homomorphism for $1 \to \mathbb{Z}_N \to \mathbb{Z}_{N^2} \to \mathbb{Z}_N \to 1$.

- $2d$ $\mathbb{Z}_2$ gauge theory with discrete theta angle given by $p$ times the quadratic refinement from the Arf invariant [15–17][5] where $p = 0, 1$, obtained by gauging a $\mathbb{Z}_2$ normal subgroup 0-form symmetry in a system with $\widetilde{G}$ 0-form symmetry. $\widetilde{G}$ is the group extension $1 \to \mathbb{Z}_2 \to \widetilde{G} \to G \to 1$. We assume the $\mathbb{Z}_2$ subgroup symmetry is non-anomalous and there is no mixed anomaly between $\mathbb{Z}_2$ and $\widetilde{G}$. If $p = 0$, the new theory does not have discrete theta angle and it has $\mathbb{Z}_2 \times G$ symmetry, while if $p = 1$ the theory has discrete theta angle given by the Arf invariant and the symmetry is the extension $\widetilde{G}$.

In Section 2, we review the global symmetry and its anomaly in $4d$ two-form gauge theory, and then study the symmetry in QFTs that couple to the two-form gauge theory. The two-form gauge theory controls the discrete theta angle of the QFTs. In Section 3, we apply the results in Section 2 to study the symmetry in $4d$ $SU(N)/\mathbb{Z}_k$ gauge theory with discrete theta angle. In Section 4, we discuss the symmetry in $4d$ gauge theory with $Spin(N), SO(N), O(N)$ gauge groups, and determine how the symmetry and its anomaly depends on the discrete theta angles. In Section 5, we review the symmetry and its anomaly in $\mathbb{Z}_N$ gauge theory, and then discuss how symmetry depends on the discrete theta angle when we gauge a $\mathbb{Z}_N$ zero-form symmetry in $3d$. In Section 6, we discuss more examples of $3d$ gauge theories with discrete theta angles and examine their global symmetry. In Section 7, we discuss gauging $\mathbb{Z}_2$ zero-form symmetry in $2d$ with or without a discrete theta angle given by the Arf invariant, and we find that the symmetry extension and 't Hooft anomaly depends on the discrete theta angle.

There are several appendices. In Appendix A we summarize some mathematical backgrounds for cochains and cohomology operations. In Appendix B we describe the symmetry extension from the analogue of the Green-Schwarz mechanism using discrete notation for discrete gauge fields. In Appendix C we discuss the symmetry in a class of TQFTs that can be defined in any spacetime dimension by generalizing the two-form gauge theory discussed

---

[5]It is the non-trivial fermionic SPT phase with unitary $\mathbb{Z}_2$ symmetry (in addition to the fermion parity) in $2d$ described by the generator in $\Omega_{\mathrm{Spin}}^2(B\mathbb{Z}_2) = \mathbb{Z}_2^2$ which is not the generator of the fermionic SPT phase without any symmetry other than the fermion parity [18,19] classified by $\Omega_{\mathrm{Spin}}^2(pt) = \mathbb{Z}_2$ [17].

in Section 2. In Appendix D we discuss gauging $\mathbb{Z}_2 \times \mathbb{Z}_2$ symmetry in $2d$ Ising $\times$ Ising model with or without discrete torsion labelled by $H^2(\mathbb{Z}_2 \times \mathbb{Z}_2, U(1)) = \mathbb{Z}_2$.

## 2 $4d$ $\mathbb{Z}_N$ two-form gauge theory

In $4d$, we can consider a two-form gauge theory with the action [20, 21, 7][6]

$$S = \int \frac{pN}{4\pi} \widehat{b} \wedge \widehat{b}, \tag{2.1}$$

where $pN$ is an even integer and $\widehat{b}$ is a $U(1)$ two-form gauge field with a constraint

$$\oint \widehat{b} \in \frac{2\pi}{N} \mathbb{Z} , \tag{2.2}$$

such that $b = \frac{N}{2\pi}\widehat{b}$ is a $\mathbb{Z}_N$ cocycle.[7] The coefficient $p$ is an integer with the identification $p \sim p + 2N$, for more details see [13]. As discussed in [21, 7, 13], the theory has a $\mathbb{Z}_N$ one-form and a $\mathbb{Z}_N$ two-form symmetry for $p = 0$. In the following we will review how the symmetries are deformed when $p$ is non-zero. Denote the background gauge fields for the higher-form symmetries by a $\mathbb{Z}_N$ 2-cochain $B_2$ and a $\mathbb{Z}_N$ 3-cocycle $Y_3$. We use the continuous notation to embed them in $U(1)$ two gauge fields $\widehat{B}_2$ and $\widehat{Y}_3$. The $\mathbb{Z}_N$ 2-cochain $B_2$ couples to the system through the following term in the action

$$\frac{N}{2\pi} \int \widehat{b} \wedge \widehat{B}_2 . \tag{2.3}$$

The $\mathbb{Z}_N$ 3-cocycle $\widehat{Y}_3$ modifies the quantization of $\widehat{b}$

$$d\widehat{b} = \widehat{Y}_3 . \tag{2.4}$$

The topological action (2.1) and the coupling (2.3) has a bulk dependence

$$\int_{5d} \frac{pN}{4\pi} d(\widehat{b} \wedge \widehat{b}) + \frac{N}{2\pi} d(\widehat{b} \wedge \widehat{B}_2) = \int_{5d} \frac{N}{2\pi} \widehat{b} \wedge (d\widehat{B}_2 + p\widehat{Y}_3) + \frac{N}{2\pi} \widehat{B}_2 \wedge \widehat{Y}_3 . \tag{2.5}$$

---

[6]See [22] for a Hamiltonian model realization of such theory.

[7]We will use variables without a hat, such as $b$, to denote a discrete gauge field and the corresponding variables without a hat, such as $\widehat{b}$, to denote its embedding in a $U(1)$ gauge field such that $\oint \widehat{b} = \frac{2\pi}{N} \oint b$. The former will be referred to as the discrete notation while the later will be referred to as the continuous notation. The subscript denotes the degree of the gauge fields. We will omit $\wedge$'s between $U(1)$ gauge fields and $\cup$'s between discrete gauge fields.

The first term involves dynamical gauge fields so it has to be removed. It can be achieved by demanding the following relation between the backgrounds

$$d\widehat{B}_2 + p\widehat{Y}_3 = 0 \ . \tag{2.6}$$

This implies that for $\gcd(N, p) \neq 1$, the background $\widehat{Y}_3$ is non-trivial while $p\widehat{Y}_3$ is trivial. When $\gcd(N, p) \neq 1$, the symmetry has an 't Hooft anomaly given by the bulk term for the background fields:

$$\frac{N}{2\pi} \int_{5d} \widehat{B}_2 \widehat{Y}_3 \ . \tag{2.7}$$

When $\gcd(N, p) = 1$, the background $\widehat{Y}_3$ is trivial and the bulk dependence (2.7) can be removed by a local counterterm $(N\alpha/4\pi) \int_{4d} \widehat{B}_2 \widehat{B}_2$ of the background fields with integer $\alpha$ that satisfies $\alpha p = 1 \bmod N$, and thus it does not represent a genuine 't Hooft anomaly.

The above computation is repeated using discrete notation in appendix B. In discrete notation, the backgrounds obey

$$\delta B_2 + p Y_3 = 0 \ , \tag{2.8}$$

and the anomaly is

$$\frac{2\pi}{N} \int_{5d} B_2 Y_3 \ . \tag{2.9}$$

## 2.1   Symmetry enrichment

The two-form gauge theory can couple to background gauge fields for other global symmetries, such as 0-form symmetries, through symmetry enrichment [23–28]. This arises naturally when the two-form gauge theory is the low energy effective theory of some ultraviolet theories. In such scenario, the ultraviolet symmetry is realized in the infrared by their actions on extended operators in the two-form gauge theory.

We can describe the coupling using the background gauge field $B_2, Y_3$ of the $\mathbb{Z}_N$ one-form and $\mathbb{Z}_N$ two-form symmetries in the two-form gauge theory that obey

$$\delta B_2 + p Y_3 = 0 \tag{2.10}$$

where $\delta$ is the coboundary operator on $C^*(M, \mathbb{Z}_N)$ for spacetime $M$.

For instance, we can couple the gauge theory to background gauge fields $X_1$ for 0-form symmetry $G^{(0)}$ and $X_2$ for one-form symmetry $G^{(1)}$ by [10, 5]

$$B_2 = X_1^* \eta_2 + f(X_2), \quad Y_3 = X_1^* \eta_3 + g(X_2) X_1^* \eta_1 + q \mathrm{Bock}(h(X_2)) \ , \tag{2.11}$$

where $q \in \mathbb{Z}_N$, $f, g, h \in \mathrm{Hom}(G^{(1)}, \mathbb{Z}_N)$, Bock is the Bockstein homomorphism for $1 \rightarrow$

$\mathbb{Z}_N \to \mathbb{Z}_{N^2} \to \mathbb{Z}_N \to 1$ and $\eta_i \in C^i(BG^{(0)}, \mathbb{Z}_N)$ are constrained such that (2.10) is satisfied.

If $\gcd(p, N) \neq 1$, the low energy TQFT has non-trivial line and surface operators obeying $\mathbb{Z}_{\gcd(p,N)}$ fusion algebra and the symmetries $G^{(0)}, G^{(1)}$ act on these operators. For instance, if $f, g, h$ is the trivial homomomorphisms and $p = 0$, non-trivial $\eta_2$ represents symmetry fractionalization for $G^{(0)}$ on the line operators. Similarly, non-trivial $\eta_3$ represents a world-volume anomaly on the surface operator charged under the two-form symmetry.

On the other hand, if $\gcd(p, N) = 1$, the two-form gauge theory is an invertible TQFT, and the symmetries $G^{(0)}, G^{(1)}$ only acts in the UV.

The symmetries $G^{(0)}, G^{(1)}$ in general has a mixed anomaly with the $\mathbb{Z}_N$ one-form symmetry depending on $p$ and $f, g, h, \eta_i$. This is given by the anomaly

$$\frac{2\pi}{N} \int_{5d} (B_2^{(0)} + B_2) Y_3 = \frac{2\pi}{N} \int_{5d} (B_2' + X_1^* \eta_2 + f(X_2)) (X_1^* \eta_3 + g(X_2) X_1^* \eta_1 + q\mathrm{Bock}(h(X_2))) \ ,$$
(2.12)

where $B_2'$ is the background for the $\mathbb{Z}_N$ one-form symmetry generated by $\exp(i \oint b)$. The mixed anomaly involving $B_2'$ is

$$\frac{2\pi}{N} \int_{5d} B_2' (X_1^* \eta_3 + g(X_2) X_1^* \eta_1 + q\mathrm{Bock}(h(X_2))) \ ,$$
(2.13)

As we will see, the mixed anomaly will be important for determining the symmetries in the theories coupled to the two-form gauge theory.

## 2.2 Couple QFT to two-form gauge theory

Suppose we start with a $4d$ theory with a non-anomalous $\mathbb{Z}_N$ one-form symmetry, and then gauge the symmetry. We have a freedom of adding an SPT phase for the $\mathbb{Z}_N$ two-form gauge fields with the action (2.1) labelled by $p$. This leads to a theory with a discrete theta angle $p$, which we will denote by $\mathcal{T}^p$. These theories are related by

$$\mathcal{T}^{k+p} \quad \longleftrightarrow \quad \frac{\mathcal{T}^k \times (\mathbb{Z}_N \text{ two-form gauge theory})_p}{\mathbb{Z}_N^{(1)}} \ .$$
(2.14)

As a special case with $k = 0$, the theories $\mathcal{T}^p$ with discrete theta angle and $\mathcal{T}^0$ without discrete theta angle are related.

Let us discuss the relations between the symmetries in $\mathcal{T}^p$ and $\mathcal{T}^0$. The symmetry in $\mathcal{T}^0$ might have a mixed anomaly with the $\mathbb{Z}_N$ one-form symmetry. When gauging the diagonal $\mathbb{Z}_N$ one-form symmetry, this contributes a non-trivial bulk dependence involving the dynamical gauge field. On the other hand, the two-form gauge theory also contributes a non-trivial bulk dependence (2.13) involving the dynamical gauge field. The two bulk

dependence cancel to give a well-defined $4d$ theory, and the cancellation might require the background gauge fields to obey certain constraints. This implies that the symmetries of $\mathcal{T}^p$ and $\mathcal{T}^0$ might be different. We will see many examples of this phenomenon in the rest of the discussions.

### 2.2.1 Gauging $\mathbb{Z}_k \subset \mathbb{Z}_N$ subgroup one-form symmetry

Let us start with a theory in $4d$ with a non-anomalous $\mathbb{Z}_N$ one-form symmetry and then gauge a $\mathbb{Z}_k$ subgroup of the symmetry. We can add an SPT phase for the $\mathbb{Z}_k$ one-form symmetry with the action (2.1) labelled by a $\mathbb{Z}_{2k}$ coefficient $p$ [13].

Let us first discuss the symmetry of the theory $\mathcal{T}^0$ with $p = 0$. The theory has an emergent $\mathbb{Z}_k$ dual one-form symmetry generated by the "Wilson surface" of the $\mathbb{Z}_k$ two-form gauge field. In addition, there is a remaining $\mathbb{Z}_N/\mathbb{Z}_k = \mathbb{Z}_{N/k}$ one-form symmetry. Denote the background two-form gauge fields for the $\mathbb{Z}_{N/k} \times \mathbb{Z}_k$ one-form symmetries by $B_2^e, B_2^m$. The two one-form symmetries have a mixed anomaly described by the $5d$ SPT phase[8]

$$\frac{2\pi}{k} \int_{5d} B_2^m \mathrm{Bock}(B_2^e) \,, \tag{2.16}$$

where Bock is the Bockenstein homomorphism of the exact sequence $1 \to \mathbb{Z}_k \to \mathbb{Z}_N \to \mathbb{Z}_{N/k}$. The anomaly arises from the symmetry extension in the original theory [6]. As a check, gauging the emergent $\mathbb{Z}_k$ dual one-form symmetry recovers the $\mathbb{Z}_N$ one-form symmetry in the original theory. Promoting the $\mathbb{Z}_k$ gauge field $B_2^m$ to a dynamical gauge field introduces an emergent $\mathbb{Z}_k$ two-form gauge field $B_2^{\prime e}$ that couples as $\frac{2\pi}{k} \int B_2^m B_2^{\prime e}$. To cancel the gauge-global anomaly (2.16), the background gauge fields obey

$$\delta B_2^{\prime e} = \mathrm{Bock}(B_2^e) \,. \tag{2.17}$$

It recovers the $\mathbb{Z}_N$ two-cocycle $k\widetilde{B}_2^{\prime e} + \widetilde{B}_2^e$ that serves as the background for the $\mathbb{Z}_N$ one-form symmetry in the original theory, where tilde denotes a lift to a $\mathbb{Z}_N$ cochain.

Next, we study the symmetry in the theory $\mathcal{T}^p$ with discrete theta angle $p$. The theory

---

[8]The anomaly has order $\gcd(k, N/k)$ *i.e.* this many copies of the system has trivial anomaly. To see this, note that there exists integers $\alpha, \beta$ such that $\gcd(N/k, k) = \alpha k + \beta N/k$. Thus multiplying the mixed anomaly by $\gcd(N/k, k)$ gives

$$\alpha \int \frac{\delta \widetilde{B}_2^e}{N/k} \widetilde{B}_2^m - \beta \int \widetilde{B}_2^e \frac{\delta \widetilde{B}_2^m}{k} = 0 \bmod 2\pi\mathbb{Z} \,, \tag{2.15}$$

where the tilde denotes a lift to a $\mathbb{Z}$ co-chain. This is consistent with the property that if $\gcd(N/k, k) = 1$, $\mathbb{Z}_N = \mathbb{Z}_{N/k} \times \mathbb{Z}_k \to 1$ so the symmetry extension $1 \to \mathbb{Z}_k \to \mathbb{Z}_N \to \mathbb{Z}_{N/k}$ is trivial.

is related to $\mathcal{T}^0$ by

$$\mathcal{T}^p \longleftrightarrow \frac{\mathcal{T}^0 \times (\mathbb{Z}_k \text{ two-form gauge theory})_p}{\mathbb{Z}_k^{(1)}} \ . \tag{2.18}$$

Gauging the diagonal $\mathbb{Z}_k$ one-form symmetry sets $B_2 = B_2'^m + b_2^m$ in (2.3) and $B_2^m = b_2^m$ where $\widehat{b}_2^m$ is the dynamical gauge field for the diagonal one-form gauge symmetry and $B_2'^m$ is the background gauge field for the residue $\mathbb{Z}_k$ one-form symmetry. The bulk dependence of the theory has two contributions from (2.7) and (2.16)

$$\frac{2\pi}{k} \int_{5d} Y_3(B_2^m + b_2^m) + b_2^m \text{Bock}(B_2^e) \ . \tag{2.19}$$

We remove the gauge-global anomaly by imposing the following constraint

$$Y_3 = \text{Bock}(B_2^e) \ . \tag{2.20}$$

The backgrounds $B_2, Y_3$ in the $\mathbb{Z}_k$ two-form gauge theory satisfy (2.10). Thus the background gauge fields $B_2^e, B_2'^m$ in $\mathcal{T}^p$ satisfy

$$\delta B_2'^m + p\text{Bock}(B_2^e) = 0 \ . \tag{2.21}$$

What's the symmetry described by such backgrounds? The relations between backgrounds can be translated into relations between symmetry charges. Denote the generators that couple to $B_2'^m$ and $B_2^e$ by $U$ and $V$ respectively, then we have the following relations

$$U^p = V^{N/k}, \quad U^k = 1 \ . \tag{2.22}$$

In particular, $U$ generates the emergent $\mathbb{Z}_k$ one-form symmetry, which is dual to the $\mathbb{Z}_k$ one-form symmetry that we gauged in the first place. For $p = 0$, $V$ generates a $\mathbb{Z}_{N/k}$ one-form symmetry and the total one-form symmetry is the direct product $\mathbb{Z}_{N/k} \times \mathbb{Z}_k$. For generic $p$, the total one-form symmetry is no longer a direct product; rather it becomes an extension of the $\mathbb{Z}_{N/k}$ one-form symmetry by the $\mathbb{Z}_k$ one-form symmetry. For instance, when $p = 1$ the total one-form symmetry is $\mathbb{Z}_N$, generated by $V$. In general, the symmetry group can be expressed as the quotient of $\mathbb{Z} \times \mathbb{Z}$ by the group generated by the columns of the following matrix

$$\begin{pmatrix} N/k & p \\ 0 & k \end{pmatrix} \ . \tag{2.23}$$

The matrix can be put into Smith normal form

$$
\begin{pmatrix} J & 0 \\ 0 & N/J \end{pmatrix} .
\tag{2.24}
$$

with $J = \gcd(k, N/k, p)$, by multiplying $SL(2, \mathbb{Z})$ matrices from the left and the right. The resulting quotient group is invariant under the transformation. Hence the one-form symmetry is

$$
\mathbb{Z}_J \times \mathbb{Z}_{N/J} .
\tag{2.25}
$$

For $p = 0$, the one-form symmetry is $\mathbb{Z}_{\gcd(N/k,k)} \times \mathbb{Z}_{N/\gcd(N/k,k)} \cong \mathbb{Z}_k \times \mathbb{Z}_{N/k}$ which reproduces the symmetry in $\mathcal{T}^0$.[9]

Substituting the relation (2.20) back to (2.19) gives the bulk dependence of the theory $\mathcal{T}^p$ that describes the 't Hooft anomaly:

$$
\frac{2\pi}{k} \int B_2'^m \mathrm{Bock}(B_2^e) .
\tag{2.28}
$$

The bulk dependence is defined up to local counterterms on the boundary. Denote

$$
L = \gcd(p, k) ,
\tag{2.29}
$$

then there exists integers $\alpha$ satisfying $\alpha p = L \bmod k$. Adding the local counterterm

$$
\frac{2\pi\alpha}{2k} \int \mathcal{P}(B_2'^m)
\tag{2.30}
$$

reduces the 't Hooft anomaly to

$$
\frac{2\pi(1 \bmod L)}{k} \int B_2'^m \mathrm{Bock}(B_2^e) .
\tag{2.31}
$$

In particular, there is no mixed anomaly for $L = 1$, but a non-trivial anomaly for $L \neq 1$. In section 3 and section 4, we will apply the above analysis to pure $SU(N)/\mathbb{Z}_k$ and $SO(N)$ gauge theory.

---

[9]The isomorphism $\mathbb{Z}_m \times \mathbb{Z}_n \cong \mathbb{Z}_{\gcd(m,n)} \times \mathbb{Z}_{mn/\gcd(m,n)}$ is as follows. Denote integers $\alpha, \beta$ that satisfy $\ell \equiv \gcd(m,n) = \alpha m + \beta n$. The element $(x, y) \in \mathbb{Z}_m \times \mathbb{Z}_n$ is mapped to

$$
x' = \alpha x + \beta y \bmod \ell, \qquad y' = -(n/\ell)x + (m/\ell)y \bmod mn/\ell .
\tag{2.26}
$$

The inverse map is

$$
x = (m/\ell)x' - \beta y' \bmod m, \qquad y' = (n/\ell)x' + \alpha y' \bmod n .
\tag{2.27}
$$

### 2.2.2 Gauging $\mathbb{Z}_k$ one-form symmetry in two-group

Let us consider gauging a $\mathbb{Z}_k$ one-form symmetry that is part of a two-group symmetry with 0-form symmetry $G$ and Postnikov class $\Theta$. Denote the background gauge field for the 0-form symmetry by $B_1$, and the gauge field for the $\mathbb{Z}_k$ one-form symmetry by $b$. The two-group symmetry implies

$$\delta b = B_1^* \Theta , \tag{2.32}$$

where $\delta$ is the differential (coboundary operator) acting on $C^*(M, \mathbb{Z}_k)$ for spacetime $M$. We can also add an SPT phase labelled by $p$ for the one-form symmetry when gauging the symmetry.

Let us begin with $p = 0$. The theory has an emergent dual $\mathbb{Z}_k$ one-form symmetry generated by $\exp(\frac{2\pi i}{k} \oint b)$, whose background gauge field we will denote by $B_2$. The two-group symmetry implies that the emergent dual $\mathbb{Z}_k$ one-form symmetry has a mixed anomaly with the 0-form symmetry $G$ [6, 10]

$$\frac{2\pi}{k} \int B_2 B_1^* \Theta . \tag{2.33}$$

Next, let us consider theory with non-zero $p$ by coupling the $p = 0$ theory to the two-form gauge theory with action labelled by $p$. From a similar analysis as in section 2.2.1, we obtain the constraint

$$Y_3 = B_1^* \Theta . \tag{2.34}$$

The constraint (2.10) in the $\mathbb{Z}_k$ two-form gauge theory implies that the backgrounds $B_2, B_1$ satisfy the constraint

$$\delta B_2 + p B_1^* \Theta = 0 . \tag{2.35}$$

Thus we find that the theory after gauging the one-form symmetry has different two-group symmetries depending on the SPT phases (labelled by $p$). More precisely, the one-form symmetry is $\mathbb{Z}_k$ and the 0-form symmetry is $G$ for all $p$, but the new Postnikov class $\Theta^{(p)}$ (that specifies how the one-form and 0-form symmetries "mix") depends on $p$ as follows

$$\Theta^{(p)} = -p\Theta . \tag{2.36}$$

The anomaly for the new two-group symmetry can be derived in a similar way as before, given by

$$\frac{2\pi(1 \bmod L)}{k} \int B_2 B_1^* \Theta , \tag{2.37}$$

where $L = \gcd(p, k)$.

# 3    $4d$ $SU(N)/\mathbf{Z}_k$ gauge theory

## 3.1    Bundle and classical action

The topology of an $SU(N)/\mathbf{Z}_k$ bundle, with $k$ a divisor of $N$, is characterized by the instanton number and the $\mathbb{Z}_k$ discrete magnetic flux $w_2^k \in H^2(M, \mathbb{Z}_k)$ where $M$ is the spacetime manifold.[10]    The magnetic flux $w_2^k$ can be understood as the obstruction to lifting the bundle to an $SU(N)$ bundle. When $w_2^k$ vanishes, the bundle can be lifted to an $SU(N)$ bundle. The instanton number is related to the magnetic flux by [12, 21, 7, 13, 30]

$$\frac{1}{8\pi^2} \int \mathrm{Tr}\,(F \wedge F) = \frac{(N - N/k)}{2k} \int \mathcal{P}(w_2^k) \mod 1 \ , \tag{3.1}$$

where $F$ is the field strength, and $\mathcal{P}(w_2^k)$ is the Pontryagin square operation (reviewed in appendix (B)).

The gauge theory can include two topological terms in the action: a continuous $\theta$ angle that multiplies the instanton number and a discrete $\mathbb{Z}_{2k}$ theta angle $p$ with $pk$ being even

$$\frac{\theta}{8\pi^2} \int \mathrm{Tr}\ F \wedge F + 2\pi \frac{p}{2k} \int \mathcal{P}(w_2^k) \ . \tag{3.2}$$

The theta angles are subjected to an identification

$$(\theta, p) \sim (\theta + 2\pi, p - (N - N/k)), \quad \text{and} \quad p \sim p + 2k \ . \tag{3.3}$$

For even $k$, the theories with discrete theta angle $p$ and $p + k$ differ by

$$\pi \int (w_2^k)^2 = \pi \int w_2^k \cup w_2(TM) \ , \tag{3.4}$$

where $w_2(TM)$ is the second Stiefel-Whitney class of the tangent bundle. Thus the only difference is that the spin of the magnetic line (with odd $\oint_{S^2} w_2^k$ on $S^2$ that surrounds the line) differs by $1/2$. If we consider gauge theory with fermions on a spin manifold (where $w_2 = 0$), then $p$ can be restricted to a $\mathbb{Z}_k$ coefficient for both $k$ even and odd, since one can modify the magnetic line by a gravitational spin $1/2$ line.

---

[10]One can also restrict the sum over instanton number, which gives rise to a modified theory with three-form symmetry [29]. We will not consider such situation in our discussion.

## 3.2 Global symmetry

The theory with discrete theta angle $p$ has the following spectrum of line operators [12].

$$W^{kq_e + pq_m} T^{q_m N/k} , \tag{3.5}$$

where $W, T$ are the basic Wilson and 't Hooft lines.

The one-form symmetry of the pure gauge theory was analyzed in appendix C of [7],

$$\mathbb{Z}_J \times \mathbb{Z}_{N/J}, \quad J = \gcd(k, N/k, p) \tag{3.6}$$

which agrees with (2.25). To relate the spectrum of line operators to the one-form symmetry in the pure gauge theory, we examine how the line operators transform under the symmetry charges. The magnetic charge $U$ in (2.22) transforms the lines as

$$\exp\left(\frac{2\pi i q_m}{k}\right) , \tag{3.7}$$

while the electric charge $V$ in (2.22) transforms the lines as

$$\exp\left(\frac{2\pi i}{N}(kq_e + pq_m)\right) . \tag{3.8}$$

We can then identify the trivial charges $U^k$ and $V^{N/k}U^{-p}$. This reproduces the relation (2.22), and thus matches the one-form symmetry (2.25). The one-form symmetry has an 't Hooft anomaly, given by (2.31). The symmetry and 't Hooft anomaly of an $SU(N)/\mathbf{Z}_k$ gauge theory have been discussed in [31]. Our results are in complete agreement.

## 3.3 Two-group symmetry in $SU(N)$ and $SU(N)/\mathbb{Z}_k$ QCD with tensor fermions

### 3.3.1 $SU(N)$ QCD

Let's consider $SU(N)$ QCD with $N_f$ Dirac fermions of equal mass in the representation $\mathbf{R}$ with $r$ boxes in its Young Tableau. For simplicity we assume that the representation $\mathbf{R}$ is complex. We first discuss the case when the fermions are massive with the same mass. The discussion is similar in the massless case. The theory has a $\mathbb{Z}_k$ one-form symmetry with $k = \gcd(N, r)$ and a flavor symmetry

$$G^{(0)} = \frac{U(N_f)}{\mathbb{Z}_{N/k}} = \frac{\widetilde{G}^{(0)}}{\mathbb{Z}_{N/k}} . \tag{3.9}$$

The fermion transforms under the group

$$\frac{\widetilde{G}^{(0)} \times SU(N)/\mathbb{Z}_k}{\mathbb{Z}_{N/k}} \ , \tag{3.10}$$

where $SU(N)$ is the gauge group, and the $\mathbb{Z}_{N/k}$ quotient identifies

$$(g, a) \sim (e^{2\pi i r/N} g, e^{-2\pi i/N} a) \in \widetilde{G}^{(0)} \times SU(N)/\mathbb{Z}_k \ . \tag{3.11}$$

Activating the background $B_2$ for the $\mathbb{Z}_k$ one-form symmetry modifies the $SU(N)$ gauge bundle to an $SU(N)/\mathbb{Z}_k$ bundle. More generally, we can simultaneously activate $B_2$ and the background $B_1$ for the $G^{(0)}$ flavor symmetry. Due to the identification (3.10), if $B_1$ is a $G^{(0)}$ background that can not be lifted to a $\widetilde{G}^{(0)}$ background, the background $B_2$ is necessarily activated. The backgrounds $B_1, B_2$ modify the gauge bundle to a $PSU(N)$ bundle [32] described by

$$w_2(PSU(N)) = \frac{N}{k}\widetilde{B_2} + \widetilde{B_1^* w_2^f} \mod N \ , \tag{3.12}$$

where $w_2(PSU(N))$ is the obstruction to lifting the gauge bundle to an $SU(N)$ bundle, tilde denotes a lift to $\mathbb{Z}_N$ cochain, and $w_2^f$ denotes the obstruction to lifting the $G^{(0)}$ background to a $\widetilde{G}^{(0)}$ background. This implies that the background satisfies a relation

$$\delta B_2 = B_1^* \text{Bock}(w_2^f) \ . \tag{3.13}$$

Thus $SU(N)$ QCD with $N_f$ massive fermions in representation $\mathbf{R}$ has a two-group symmetry that combines the $\mathbb{Z}_k$ one-form symmetry and the flavor symmetry $G^{(0)}$, as described by the Postnikov class

$$\Theta^{SU(N)} = \text{Bock}(w_2^f) \ . \tag{3.14}$$

When the fermions are massless, the flavor symmetry $G^{(0)}$ is enlarged to

$$G^{(0)} = \frac{\widetilde{G}^{(0)}}{\mathbb{Z}_{N/k}}, \quad \widetilde{G}^{(0)} = \left( \frac{SU(N_f)_L \times SU(N_f)_R \times U(1) \times \mathbb{Z}_{2N_f I(\mathbf{R})}}{\mathbb{Z}_2} \times \mathbb{Z}_{N_f} \times \mathbb{Z}_{N_f} \right) \ . \tag{3.15}$$

Here $I(\mathbf{R})$ is the index of the representation $\mathbf{R}$.[11] The quotient in $\widetilde{G}^{(0)}$ introduces the

---

[11]The index is defined as $\text{Tr}_{\mathbf{R}}(T^a T^b) = \delta^{ab} I(\mathbf{R})/(2h_G^\vee)$ with $h_G^\vee$ the dual Coxeter number and $T^a$ normalized such that $\text{Tr}_{\mathbf{adj}}(T^a T^b) = \delta^{ab}$. The index for the fundamental representation and the adjoint representation is 1 and $2N$ respectively.

following identification

$$
\begin{aligned}
(g, h, b, c) &\sim (g, h, -b, -c) \\
&\sim (e^{2\pi i/N_f} g, e^{2\pi i/N_f} h, e^{-2\pi i/N_f} b, c) \\
&\sim (e^{2\pi i/N_f} g, e^{-2\pi i/N_f} h, b, e^{2\pi i/N_f} c) \\
&\in SU(N_f)_L \times SU(N_f)_R \times U(1) \times \mathbb{Z}_{2N_f I(\mathcal{R})} \ .
\end{aligned}
\tag{3.16}
$$

The fermion transforms under the group

$$
\frac{\widetilde{G}^{(0)} \times SU(N)/\mathbb{Z}_k}{\mathbb{Z}_{N/k}} \ ,
\tag{3.17}
$$

where $SU(N)$ is the gauge group and the $\mathbb{Z}_{N/k}$ quotient identifies

$$
(a, b) \sim (e^{2\pi i/N} a, e^{-2\pi i r/N} c) \in SU(N)/\mathbb{Z}_k \times U(1) \ .
\tag{3.18}
$$

Similar to the massive theory, the massless QCD also has a two-group symmetry that combines the $\mathbb{Z}_k$ one-form symmetry and the enlarged flavor symmetry $G^{(0)}$ in (3.15). The corresponding Postnikov class is still given by (3.14) with $w_2^f$ now being the obstruction to lifting the $G^{(0)}$ background to a $\widetilde{G}^{(0)}$ background with $G^{(0)}, \widetilde{G}^{(0)}$ in (3.15).

### 3.3.2 $SU(N)/\mathbf{Z}_k$ QCD

Next, we gauge the $\mathbb{Z}_k$ one-form symmetry with an SPT phase labelled by $p$ for the $\mathbb{Z}_k$ two-form gauge field. This turns the theory into an $SU(N)/\mathbf{Z}_k$ gauge theory with discrete theta angle $p$ that couples to $N_f$ Dirac fermions in representation $\mathbf{R}$. The theory has the $G^{(0)}$ symmetry (3.9) and a magnetic $\mathbb{Z}_k$ one-form symmetry whose background is denoted by $B^m$.

Following the discussion in section 2.2.2, the background gauge fields satisfy a constraint

$$
\delta B^m = p B_1^* \text{Bock}(w_2^f) \ ,
\tag{3.19}
$$

which describes a two-group symmetry that combines the $\mathbb{Z}_k$ one-form symmetry and the flavor symmetry $G^{(0)}$, with Postnikov class that depends on the discrete theta angle

$$
\Theta^{SU(N)/\mathbb{Z}_k} = p \text{Bock}(w_2^f) \ .
\tag{3.20}
$$

The two-group symmetry has a mixed anomaly

$$
\frac{2\pi (1 \bmod L)}{k} \int B^m B_1^* \text{Bock}(w_2^f) \ ,
\tag{3.21}
$$

where $L = \gcd(p, k)$. In particular, there is no mixed anomaly if $L = 1$ but a non-trivial anomaly otherwise.

When the fermion mass is large, the theory flows to an $SU(N)/\mathbf{Z}_k$ pure gauge theory with an accidental electric one-form symmetry. The two-group symmetry and the anomaly is realized by the symmetry enrichment $B^e = B_1^* w_2^f$. In the ultraviolet theory, one can interpret the relation as an explicit breaking of the center one-form symmetry by the screening from the fermion fields.

# 4   $4d$ gauge theories with $\mathfrak{so}(N)$ Lie algebra

In this section, we will discuss $4d$ gauge theories associated with $\mathfrak{so}(N)$ Lie algebra, including $Spin(N), SO(N)$ and $O(N)$ gauge theories.

## 4.1   Bundle

The gauge bundles associated with $\mathfrak{so}(N)$ Lie algebra were reviewed in [3]. Here we briefly summarize their topology which are characterised by the Stiefel-Whitney characteristic classes $w_i \in H^i(M, \mathbb{Z}_2)$, where $M$ is the spacetime four-manifold.

The $O(N)$ group has the largest set of possible bundles. They are characterized by $w_1$ and $w_2^{(1)}$, where $w_1$ is the obstruction to restricting the bundle to an $SO(N)$ bundle while $w_2^{(1)}$ is the obstruction to lifting the bundle to a $Pin^+(N)$ bundle. All the $SO(N)$ bundles can be constructed by restricting the $O(N)$ bundles whose $w_1$ vanishes. The $SO(N)$ bundles are then characterized by $w_2^{(1)}$, which is the obstruction to lifting the bundle to an $Spin(N)$ bundle. All the $Spin(N)$ bundle can be constructed by lifting the $SO(N)$ bundles whose $w_2^{(1)}$ vanishes.

For even $N$, we can also consider $PSO(N) = SO(N)/\mathbb{Z}_2$ or $PO(N) = O(N)/\mathbb{Z}_2$ bundles. The $PO(N)$ bundles have another characteristic class $w_2^{(2)} \in H^2(M, \mathbb{Z}_2)$, which is the obstruction to lifting the bundles to $O(N)$ bundles. The characteristic classes $w_2^{(1)}, w_2^{(2)}, w_1$ obey certain constraint

$$\delta w_2^{(1)} = \frac{N}{2}\text{Bock}(w_2^{(2)}) + w_2^{(2)}w_1, \quad \delta w_2^{(2)} = 0, \quad \delta w_1 = 0 , \tag{4.1}$$

where Bock is the Bockstein homomorphism associated with the extension $1 \to \mathbb{Z}_2 \to \mathbb{Z}_4 \to \mathbb{Z}_2 \to 1$. The constraint can be understood from the properties of the $Spin(N)$ group.

For even $N$, $Spin(N)$ has a center of order 4 ($\mathbb{Z}_2 \times \mathbb{Z}_2$ for $N = 0$ mod 4 and $\mathbb{Z}_4$ for $N = 2$ mod 4). The obstruction to lifting a $PO(N)$ bundle to an $Spin(N)$ bundle is then characterized by $w_2^{(1)}, w_2^{(2)} \in H^2(M, \mathbb{Z}_2 \times \mathbb{Z}_2)$ for $N = 0$ mod 4, and $\widetilde{w_2}^{(2)} + 2\widetilde{w_2}^{(1)} \in$

$H^2(M, \mathbb{Z}_4)$ for $N = 2 \bmod 4$, where tilde denotes the lift to a $\mathbb{Z}_4$ cochain. This leads to the first term in the constraint (4.1).

The $Spin(N)$ group has a $\mathbb{Z}_2$ charge conjugation outer-automorphism. It acts non-trivially on the center of $Spin(N)$ for even $N$. To see this, we can consider the semi-direct product group $Pin^+(N) = Spin(N) \rtimes \mathbb{Z}_2$. Its center is $\mathbb{Z}_2$ for even $N$ [33–36]. It implies that the charge conjugation outer-automorphism acts non-trivially on the center of $Spin(N)$ leaving a $\mathbb{Z}_2$ subgroup invariant. This leads the second term in the constraint (4.1).

Without loss of generality, we will always assume even $N$ in the rest of the section unless specified. The discussions can be applied to the odd $N$ cases simply by setting $w_2^{(2)}$ and other relevant quantities to be zero.

## 4.2 Classical action

### 4.2.1 Continuous theta angle

Both $Spin(N)$ and $SO(N)$ gauge theories have continuous $\theta$ angle that multiplies the instanton number. In $Spin(N)$ gauge theory it is $2\pi$ periodic, while in $SO(N)$ gauge theory with $N > 3$ it is $4\pi$ periodic on a non-spin manifold but $2\pi$ periodic on a spin manifold [30].[12] The difference between $\theta$ and $\theta + 2\pi$ is that the basic 't Hooft line in the $SO(N)$ gauge theory differs in their spin by $1/2$, and thus they are indistinguishable on spin manifolds (or in a fermionic theory) where the theory has gravitational fermion line that can be used to modify the line operators without changing the dynamics.

### 4.2.2 Discrete theta angles

$Spin(N)$ gauge theory does not have any discrete theta angle while $SO(N)$ gauge theory admits discrete theta angle

$$2\pi \frac{p}{4} \int \mathcal{P}(w_2^{(1)}) \, , \tag{4.2}$$

where $p$ is a $\mathbb{Z}_4$ coefficient. The theta angles are subject to the identification [30]

$$
\begin{aligned}
N = 3: &\quad (\theta, p) \sim (\theta + 2\pi, p + 1), \quad \text{and} \quad p \sim p + 4 \, , \\
N > 3: &\quad (\theta, p) \sim (\theta + 2\pi, p + 2), \quad \text{and} \quad p \sim p + 4 \, .
\end{aligned}
\tag{4.3}
$$

The theories with $p$ and $p + 2$ differ in the spin of the basic 't Hooft lines.[13] Hence they are indistinguishable on spin manifolds. As discussed in [12], the theories with $p = 0$, denoted

---

[12]When $N = 3$, the theta angle in the $SO(3)$ gauge theory is $4\pi$ periodic on a spin manifold and $8\pi$ periodic on a non-spin manifold.

[13]This follows from the identity $\pi \int w_2^{(1)} \cup w_2^{(1)} = \pi \int w_2^{(1)} \cup w_2(TM)$ where $w_2(TM)$ is the second Stiefel-Whitney class of the tangent bundle.

by $SO(N)_+$ and $p = 1$, denoted by $SO(N)_-$ have different line operator spectrum.

$O(N)$ gauge theory has an additional discrete theta angle given by

$$r\pi \int w_2^{(1)} \cup (w_1)^2 = r\pi \int w_2^{(1)} \cup \text{Bock}(w_1) \ , \tag{4.4}$$

where $r$ is a $\mathbb{Z}_2$ coefficient and Bock is the Bockstein homomorphism for $1 \to \mathbb{Z}_2 \to \mathbb{Z}_4 \to \mathbb{Z}_2 \to 1$. This is an analogue of the topological term $\pi \int w_1 w_2$ in $3d$ $O(N)$ gauge theory discussed in [37, 3].[14]

## 4.3   Global symmetry

### 4.3.1   $Spin(N)$ gauge theory

The theory has an electric one-form symmetry $\mathcal{A}$ determined by the center of the gauge group:

$$\mathcal{A} = \begin{cases} \mathbb{Z}_2 & \text{odd } N \\ \mathbb{Z}_4 & N = 2 \text{ mod } 4 \\ \mathbb{Z}_2 \times \mathbb{Z}_2 & N = 0 \text{ mod } 4 \end{cases} . \tag{4.5}$$

Denote the background for the $\mathbb{Z}_2$ subgroup one-form symmetry by $B_2^{(1)}$. For even $N$, the one-form symmetry includes an additional $\mathbb{Z}_2$ factor whose background is denoted by $B_2^{(2)}$. The theory also has a $\mathbb{Z}_2$ 0-form charge conjugation symmetry $\mathcal{C}$ whose background is denoted by $B_1^{\mathcal{C}}$.

Activating only the charge conjugation background twists the gauge bundle to a $Pin^+(N)$ bundle. More generally, we can activate all these background which twists the gauge bundle to a $PO(N)$ bundle with the characteristic classes

$$w_1 = B_1^{\mathcal{C}}, \quad w_2^{(1)} = B_2^{(1)}, \quad w_2^{(2)} = B_2^{(2)} \ . \tag{4.6}$$

The constraint (4.1) then implies the following relation

$$\delta B_2^{(1)} = \frac{N}{2} \text{Bock}\left(B_2^{(2)}\right) + B_2^{(2)} B_1^{\mathcal{C}} \ . \tag{4.7}$$

We can also couple the theory to a different $\mathbb{Z}_2$ background gauge field $B_1$ through a non-trivial symmetry fractionalization

$$B_1^{\mathcal{C}} = B_1, \quad B_2^{(1)} = B_1 \cup B_1, \quad B_2^{(2)} = 0. \tag{4.8}$$

---

[14]On orientable manifold, $\pi \int w_3 w_1 = \pi \int \left(\text{Bock}(w_2^{(1)})w_1 + w_2^{(1)}(w_1)^2\right) = \pi \int Sq^1(w_2^{(1)}w_1) = 0 \text{ mod } 2\pi\mathbb{Z}$, and thus the $\pi \int w_3 w_1$ term is trivial.

In such case, the lines charged under the $\mathbb{Z}_2$ one-form symmetry that couples $B_2^{(1)}$, such as the Wilson lines in the spinor representations, are in the projective representation of the $\mathbb{Z}_2$ symmetry whose generator becomes of order 4. Activating this $\mathbb{Z}_2$ background twists the gauge bundle to a $Pin^-$ bundle with $w_2^{(1)} = w_1 \cup w_1$. This is consistent with the properties of the $Pin^-(N)$ bundles [16].

All of these global symmetries do not have 't Hooft anomalies *i.e.* they can be gauged. Gauging the charge conjugation symmetry extends the gauge group to $Pin^+(N)$ while gauging the $\mathbb{Z}_2$ symmetry with non-trivial fractionalization extends the gauge group to $Pin^-(N)$.

### 4.3.2 $SO(N)$ gauge theory

$SO(N)$ gauge theory can be constructed from $Spin(N)$ gauge theory by gauging the $\mathbb{Z}_2$ subgroup one-form symmetry that does not transform the Wilson lines in the vector representations (as opposite to Wilson lines in the spinor representations). For $N = 2$ mod 4, it gauges the $\mathbb{Z}_2$ subgroup of the $\mathbb{Z}_4$ one-form symmetry while for $N = 0$ mod 4, it gauges one of the $\mathbb{Z}_2$'s of the $\mathbb{Z}_2 \times \mathbb{Z}_2$ symmetry.

The theory has a $\mathbb{Z}_4$ discrete theta angle $p$. The theories with $p$ and $p + 2$ are related by the coupling $\pi \int w_2^{(1)} \cup w_2$ which shifts the spin of the basic 't Hooft line by $1/2$.

The theory has an emergent dual $\mathbb{Z}_2$ magnetic one-form symmetry whose background is denoted by $B_2^m$. The full one-form symmetry depends on the discrete theta angle, following from (2.25). It is summarized in table 1. The theory still has the $\mathbb{Z}_2$ charge conjugation symmetry whose background is $B_1^{\mathcal{C}}$. It acts as a $\mathbb{Z}_2$ outer-automorphism on the one-form symmetry when $N$ is even. For even $N$ the background gauge fields satisfy

$$\delta B_2^m = p \left( \frac{N}{2} \text{Bock}(B_2^{(2)}) + B_2^{(2)} B_1^{\mathcal{C}} \right) , \tag{4.9}$$

where $B_2^{(2)}$ is the background for the remaining electric one-form symmetry. The backgrounds are independent when $p$ is even, but are correlated when $p$ is odd. In particular, activating $B_2^{(2)}$ and $B_1^{\mathcal{C}}$ necessarily activates $B_2^m$.

Let us discuss the 't Hooft anomaly of these symmetries. For odd $N$ there is no 't Hooft anomaly. For even $N$, the anomaly depends on the discrete theta angle $p$

- When $p = 0$, the symmetries have an 't Hooft anomaly. The relation (4.1) implies that in the presence of the backgrounds $B_2^{(2)}$ and $B_1^{\mathcal{C}}$, the coupling to the background $B_2^m$ for the magnetic one-form symmetry is not well-defined: it depends on the bulk by

$$\pi \int \delta w_2^{(1)} B_2^m = \pi \int \left( \frac{N}{2} \text{Bock}(B_2^{(2)}) + B_2^{(2)} B_1^{\mathcal{C}} \right) B_2^m . \tag{4.10}$$

| | odd $N$ | $N = 2$ mod 4 | $N = 0$ mod 4 |
|---|---|---|---|
| even $p$ | $\mathbb{Z}_2$ | $\mathbb{Z}_2 \times \mathbb{Z}_2$ | $\mathbb{Z}_2 \times \mathbb{Z}_2$ |
| odd $p$ | $\mathbb{Z}_2$ | $\mathbb{Z}_4$ | $\mathbb{Z}_2 \times \mathbb{Z}_2$ |

Table 1: One-form symmetry in $SO(N)$ pure gauge theory with discrete theta angle $p$.

The theory flows to the $\mathbb{Z}_2$ two-form gauge theory (2.1) with $p = 0$ in the infrared [12]. The anomaly is matched by the symmetry fractionalization map $Y_3 = \frac{N}{2}\text{Bock}(B_2^{(1)}) + B_2^{(1)}B_1^{\mathcal{C}}$, $B_2 = B_2^m$ for $Y_3$ and $B_2$ in (2.7)

- When $p = 1$, the symmetries have no 't Hooft anomaly. The anomaly (4.10) can be removed by adding the local counterterm $\frac{\pi}{2}\int \mathcal{P}(B_2^m)$. The topological terms in the action

$$\frac{\pi}{2}\int \mathcal{P}(w_2^{(1)}) + \pi \int w_2^{(1)} B_2^m + \frac{\pi}{2}\int \mathcal{P}(B_2^m) = \frac{\pi}{2}\int \mathcal{P}(w_2^{(1)} + B_2^m) \ , \qquad (4.11)$$

then becomes well-defined since $w_2^{SO} + B^m$ is a $\mathbb{Z}_2$ two-cocycle.

Let us now discuss gauging the $\mathbb{Z}_2$ magnetic one-form symmetry in the $SO(N)$ gauge theory. The gauging promotes the gauge field $B_2^m$ to a dynamical gauge field $b_2^m$. We can introduce a new $\mathbb{Z}_2$ two-form background gauge field $B_2'$ that couples to the theory as $\pi \int b_2^m B_2'$. We again have the freedom to include an additional SPT phase, $\frac{p'\pi}{2}\int \mathcal{P}(b_2^m)$ with $p' = 0, 1$:

- $p' = 0$. The gauge field $b_2^m$ acts as a Lagrangian multiplier that forces $w_2^{(1)} = 0$ thus we recover the $Spin(N)$ gauge theory. To cancel the bulk dependence (4.10), the background fields are constrained such that

$$\delta B_2' = \frac{N}{2}\text{Bock}(B_2^{(2)}) + B_2^{(2)}B_1^{\mathcal{C}} \ . \qquad (4.12)$$

This recovers the constraint (4.7) on the background fields in $Spin(N)$ gauge theory if we identified $B_2'$ with $B_2^{(1)}$. Hence the full symmetry in $Spin(N)$ gauge theory is recovered. The discussion also applies to the $p = 1$ case since the local counterterm for $b_2^m$ is fixed and no longer can be used to cancel the bulk dependence.

When $p = 1$, the symmetry extension (4.9) in the $SO(N)$ gauge theory implies that the coupling $\pi \int b^m B_2'$ is not well-defined but has a bulk dependence

$$\pi \int \delta b^m B_2' = \pi \int \left( \frac{N}{2}\text{Bock}(B_2^{(2)}) + B_2^{(2)}B_1^{\mathcal{C}} \right) B_2' = \pi \int \delta B_2' B_2' \ , \qquad (4.13)$$

which can be cancelled by the classical local counterterm $(\pi/2)\int \mathcal{P}(B_2')$.

We conclude that gauging the $\mathbb{Z}_2$ magnetic one-form symmetry with $p' = 0$ recovers the original non-anomalous symmetries of the $Spin(N)$ gauge theory.

- $p' = 1$. After gauging, the resulting theory is equivalent to an $SO(N)$ gauge theory with a shifted discrete theta angle $p \to p - 1$. This follows from the fact that the TQFT for $b_2^m$ with $p' = 1$ is invertible so we can integrate out $b_2^m$ as

$$\frac{p\pi}{2} \int \mathcal{P}(w_2^{(1)}) + \pi \int w_2^{(1)} b_2^m + \frac{\pi}{2} \int \mathcal{P}(b_2^m) = \frac{(p-1)\pi}{2} \int \mathcal{P}(w_2^{(1)}) + \frac{\pi}{2} \int \mathcal{P}(b_2'^m) , \quad (4.14)$$

where the last term is a decoupled invertible TQFT of a $\mathbb{Z}_2$ two-form gauge field $b_2'^m = b_2^m + w_2^{(1)}$.

Let us now show how the symmetry and anomaly for theories with different $p$ are related by such gauging. If we start with the $p = 0$ theory, $\delta b_2^m = 0$ so the action

$$\pi \int (w_2^{(1)} + B_2') b_2^m + \frac{\pi}{2} \int \mathcal{P}(b_2^m) \quad (4.15)$$

is well-defined only if the combination $w_2^{(1)} + B_2'$ is closed. This imposes the following constraints on the background

$$\delta B_2' = -\delta w_2^{(1)} = \frac{N}{2} \mathrm{Bock}(B_2^{(2)}) + B_2^{(2)} B_1^{\mathcal{C}} \mod 2 , \quad (4.16)$$

which agrees with the symmetry of the $SO(N)$ theory with odd $p$ if we identified $B_2'$ with $B_2^{(1)}$.

On the other hand, if we start with the $p = 1$ theory,

$$\delta b_2^m = \delta w_2^{(1)} = \frac{N}{2} \mathrm{Bock}(B_2^{(1)}) + B_2^{(1)} B_1^{\mathcal{C}} , \quad (4.17)$$

so the action

$$\pi \int B_2' b_2^m + \frac{\pi}{2} \int \mathcal{P}(b_2^m + w_2^{(1)}) \quad (4.18)$$

is well-defined if $\delta B_2' = 0$. The action has a bulk dependence

$$\pi \int \left( \frac{N}{2} \mathrm{Bock}(B_2^{(2)}) + B_2^{(2)} B_1^{\mathcal{C}} \right) B_2' , \quad (4.19)$$

which agrees with the 't Hooft anomaly of the $SO(N)$ gauge theory with even $p$ if we identified $B_2'$ with $B_2^{(1)}$.

We conclude that gauging the $\mathbb{Z}_2$ magnetic one-form symmetry with $p' = 1$ reproduces the symmetry and anomaly of the $SO(N)$ gauge theory with a shifted discrete theta

angle $p \rightarrow p - 1$.

### 4.3.3 $O(N)$ gauge theory

The $O(N)$ gauge theory can be constructed from the $SO(N)$ gauge theory by gauging the $\mathbb{Z}_2$ charge conjugation symmetry. The theory has the following symmetries

- $\mathbb{Z}_2$ magnetic one-form symmetry generated by $\exp(i\pi \oint w_2^{(1)})$. Denote its background by $B_2^m$.

- $\mathbb{Z}_2$ two-form symmetry generated by $\exp(i\pi \oint w_1)$. Denote its background by $B_3$.

- $\mathbb{Z}_2 \times \mathbb{Z}_2$ 0-form symmetries generated by $\exp(i\pi \oint w_1 w_2^{(1)}), \exp(i\pi \oint w_1^3)$ [5]. Denote the corresponding backgrounds by $X_1, Y_1$.

These symmetries are free of 't Hooft anomaly and can be gauged.

If $N$ is even, the theory has an additional electric one-form symmetry. Denote its background by $B_2^{(2)}$. Depending on the discrete theta angles $(p, r)$ in (4.2)(4.4), the center one-form symmetry can form different symmetry groups with the other symmetries and they can have non-trivial 't Hooft anomaly as discussed below. We will always assume that $N$ is even.

**(p,r)=(0,0): no discrete theta angles.** The electric one-form symmetry is $\mathbb{Z}_2$. Its background $B_2^{(2)}$ modifies

$$\delta w_2^{(1)} = \frac{N}{2}\text{Bock}(B_2^{(2)}) + B_2^{(2)} w_1 . \tag{4.20}$$

Thus the coupling

$$\pi \int w_2^{(1)} B_2^m + \pi \int w_1 w_2^{(1)} X_1 , \tag{4.21}$$

is no-longer well-defined. We can extend the fields to the bulk, then these terms have the bulk dependence

$$\pi \int \left( \frac{N}{2}\text{Bock}(B_2^{(2)}) + B_2^{(2)} w_1 \right) (B_2^m + w_1 X_1) , \tag{4.22}$$

where the part that represents a gauge-global anomaly can be cancelled by $\pi \int w_1 B_3$ with the modified cocycle condition

$$\delta B_3 = B_2^e B_2^m + \frac{N+2}{2}\text{Bock}(B_2^e) X_1 + B_2^e \text{Bock}(X_1) . \tag{4.23}$$

This can be interpreted as a three-group symmetry. The classical term in the bulk is the SPT phase that describes the 't Hooft anomaly for the 3-group symmetry:

$$\frac{N}{2} \int \text{Bock}(B_2^{(2)}) B_2^m \ . \tag{4.24}$$

**(p,r)=(0,1).** In addition to the bulk dependence (4.22), the topological term of $r = 1$ also contributes to the bulk dependence

$$\pi \int (w_1)^2 \delta w_2^{(1)} = \frac{N}{2} \pi \int w_1(TM) w_1 \text{Bock}(B_2^{(2)}) + \pi \int B_2^{(2)} (w_1)^3 \ , \tag{4.25}$$

where the first term is trivial on orientable manifolds (whose first Stiefel-Whitney class $w_1$ vanishes). The last term represents a gauge-global anomaly. One can attempt to cancel it using the coupling $\pi \int Y_1(w_1)^3$, which leads to the relation

$$B_2^{(2)} = \delta Y_1 \ . \tag{4.26}$$

It implies that the background $B_2^{(2)}$ is trivial. Hence the electric one-form symmetry is explicitly broken. Another way to see this is by examining the one-form symmetry transformation $B_2^{(2)} \to B_2^{(2)} + \delta\lambda$, which shifts $w_2^{(1)} \to w_2^{(1)} + w_1\lambda$ due to the constraint (4.20), and accordingly transforms the topological term of $r = 1$ by

$$\pi \int (w_1^O)^3 \lambda \ . \tag{4.27}$$

This implies that the one-form symmetry transformation is broken by the point operators which carry non-trivial flux $\exp(i\pi \oint_{S^3}(w_1)^3) = -1$ on the $S^3$ surrounding it.[15]

Thus in contrast to the case $(p, r) = (0, 0)$, the symmetries do not form a three-group, and the anomaly (4.24) vanishes since $B_2^{(2)}$ vanishes.

**(p, r) = (1, 0).** In additional to the bulk dependence (4.22), the topological term of $p = 1$ also contributes to the bulk dependence

$$\pi \int w_2^{(1)} \left( \frac{N}{2} \text{Bock}(B_2^{(2)}) + B_2^{(2)} w_1 \right) \tag{4.28}$$

---

[15] As discussed in section 4.3 of [6], this anomalous transformation (4.27) can be interpreted as a gauge anomaly $\pi \int_{3d}(w_1)^3$ on the worldvolume of the one-form symmetry defect (which is a surface operator). This gauge anomaly can be cancelled by introducing a non-trivial TQFT coupled to $w_1$ on the worldvolume of the surface operator, which gives rise to non-invertible defects.

It represents a gauge-global anomaly. One can attempt to cancel the second term using the coupling $\pi \int w_2^{(1)} w_1 X_1$, which leads to the relation

$$B_2^{(2)} = \delta X_1 . \tag{4.29}$$

It implies that the background $B_2^{(2)}$ is trivial. Hence the electric one-form symmetry is explicitly broken. It is violated by the point operators that carry non-trivial flux $\exp(i\pi \int_{S^3} w_1 w_2^{(1)}) = -1$ on the $S^3$ that surrounds it.

Thus in contrast to the previous case $(p, r) = (0, 0)$, the symmetries do not form a three-group, and the anomaly (4.24) vanishes since $B_2^{(2)}$ vanishes.

$(\mathbf{p}, \mathbf{r}) = (\mathbf{1}, \mathbf{1})$.    The theory has three contributions (4.22), (4.25), (4.28) to its bulk dependence. The gauge-global anomaly implies that the electric one-form symmetry is broken explicitly by the operator with non-trivial flux $\exp\left(i\pi \int_{S^3}(w_1 w_2^{(1)} + (w_1)^3)\right) = -1$ on the $S^3$ surrounding it.

Thus in contrast to the case $(p, r) = (0, 0)$, the symmetries do not form a three-group, and the anomaly (4.24) vanishes since $B_2^{(2)}$ vanishes.

## 4.4    Two-group symmetry in $Spin(N)$ QCD with vector fermions

Consider $Spin(N)$ gauge theory with $N_f$ massless Weyl fermions in the vector representation. The theory has mesons $\psi_I^a \psi_J^a$ and baryons $\epsilon_{a_1 \cdots a_N} \psi_{I_1}^{a_1} \cdots \psi_{I_N}^{a_N}$ as local operators where $I, J = 1, \cdots, N_f$ and $a_1, a_2 \cdots = 1, \cdots N$ are flavor and color indices.

The baryons are charged under a $\mathbb{Z}_2$ charge conjugation symmetry. We will denote the symmetry by $\mathbb{Z}_2^{\mathcal{C}}$ and denote its background by $B_1^{\mathcal{C}}$. Two baryons can annihilate into mesons using the identity

$$\epsilon_{a_1 \cdots a_N} \epsilon_{a_1' \cdots a_N'} = \sum_{\sigma} (-1)^{\text{sign}(\sigma)} \delta_{a_1 a_{\sigma(1)}'} \cdots \delta_{a_N a_{\sigma(N)}'} . \tag{4.30}$$

Hence the baryon number is only conserved mod 2 so it can be identified with the $\mathbb{Z}_2$ charge of the charge conjugation symmetry.

The baryons and mesons transform under an $\widetilde{SU}(N_f) = (SU(N_f) \times \mathbb{Z}_{2N_f})/\mathbb{Z}_{N_f}$ flavor symmetry. When $N_f$ is odd, the flavor symmetry factorizes into $\widetilde{SU}(N_f) = SU(N_f) \times \mathbb{Z}_2$. When $N$ is even, the $(-1)^F$ symmetry, which is a $\mathbb{Z}_2$ subgroup of the flavor symmetry, can be identified with a gauge rotation in the center of the gauge group and thus acts trivially on the local operators. When $N$ is odd, the $(-1)^F$ symmetry is identified with the charge

conjugation symmetry. In summary, the ordinary global symmetry is

|  | Odd $N$ | Even $N$ |
|---|---|---|
| Odd $N_f$ | $SU(N_f) \times \mathbb{Z}_2$ | $SU(N_f) \rtimes \mathbb{Z}_2^{\mathcal{C}}$ |
| Even $N_f$ | $\widetilde{SU}(N_f)$ | $\left(\widetilde{SU}(N_f)/\mathbb{Z}_2\right) \rtimes \mathbb{Z}_2^{\mathcal{C}}$ |

$$(4.31)$$

We will focus on the cases when both $N$ and $N_f$ are even. The background for the flavor symmetry can be decomposed into a $PSU(N_f)$ gauge field $A$ and a $\mathbb{Z}_{2N_f}$ gauge field $A_\chi$ with a constraint

$$\delta A_\chi = 2\widetilde{w}_2^{(N_f)} + N_f \widetilde{w}_2^f \mod 2N_f , \qquad (4.32)$$

where $w_2^{(N_f)}$ is the obstruction to lifting the $PSU(N_f)$ bundle to an $SU(N_f)$ bundle, and $w_2^f$ is the obstruction to lifting the $\widetilde{SU}(N_f)/\mathbb{Z}_2$ bundle to an $\widetilde{SU}(N_f)$ bundle.

The theory also has Wilson lines in the spinor representations that are not screened by the matter. The Wilson lines are charged under a $\mathbb{Z}_2$ electric one-form symmetry. We will denote the background for the one-form symmetry by $B_2^{(2)}$. For even $N$ and $N_f$, the one-form symmetry combines with the flavor symmetry to a two-group symmetry.[16] The background of the two group symmetry is

$$\delta B_2^{(2)} = \frac{N}{2}\text{Bock}(w_2^f) + w_2^f B_1^{\mathcal{C}} . \qquad (4.33)$$

The one-form symmetry is expected to be unbroken at low energy which signals confinement. The two-group symmetry implies that the strings charged under the one-form symmetry carry an 't Hooft anomaly on their worldsheet characterized by

$$\pi \int_{3d} \left(\frac{N}{2}\text{Bock}(w_2^f) + w_2^f B_1^{\mathcal{C}}\right) . \qquad (4.34)$$

## 4.5 Two-group symmetry in $SO(N)$ QCD with $N_f$ vector fermions

The theory can be constructed from $Spin(N)$ QCD by gauging the $\mathbb{Z}_2$ electric one-form symmetry. One can include discrete theta angle $p$. As in $Spin(N)$ QCD, We will focus on the case where both $N, N_f$ are even.

The theory has a dual $\mathbb{Z}_2$ magnetic one-form symmetry generated by $\exp(i\pi \oint w_2^{(1)})$. Denote the background for the dual magnetic one-form symmetry by $B_2^m$, which couples to the theory as

$$\pi \int w_2^{(1)} B_2^m . \qquad (4.35)$$

---

[16]Similar results on the two-group symmetries (without the charge conjugation symmetry) in $Spin(N)$ and $SO(N)$ QCD have been obtained independently by Yasunori Lee, Kantaro Ohmori and Yuji Tachikawa.

Since $\pi \int w_2^{(1)} w_2^{(1)} = \pi \int w_2^{(1)} w_2$ with $w_2$ the second Stiefel-Whitney class of the space-time manifold, we have the identification $(p, B_2^m) \sim (p + 2, B_2^m + w_2)$. Without loss of generality, we can restrict to $p = 0, 1$. These two theories are denoted by $SO(N)_+$ and $SO(N)_-$ respectively.

### 4.5.1 $SO(N)_+$ QCD

When $p = 0$, the two-group symmetry in the $Spin(N)$ QCD becomes a mixed anomaly after gauging

$$\pi \int \delta w_2^{(1)} \cup B_2^m = \pi \int \left( \frac{N}{2} \text{Bock}(w_2^f) + w_2^f B_1^{\mathcal{C}} \right) B_2^m . \tag{4.36}$$

It is a mixed anomaly between the magnetic one-form symmetry and the flavor symmetry (and charge conjugation symmetry).

### 4.5.2 $SO(N)_-$ QCD

When $p = 1$, the theory couples to the two-form gauge theory (2.1). Applying the discussion in section 2.2.2, the theory has a two-group symmetry whose backgrounds obey the relation

$$\delta B_2^m = \frac{N}{2} \text{Bock}(w_2^f) + w_2^f B_1^{\mathcal{C}} . \tag{4.37}$$

In contrast to $SO(N)_+$ QCD, the symmetry has no 't Hooft anomaly.

# 5 $3d$ $\mathbb{Z}_N$ one-form gauge theory

## 5.1 Bosonic $\mathbb{Z}_N$ one-form gauge theory

In $3d$, we can consider a class of $\mathbb{Z}_N$ one-form gauge theories that can be constructed from $U(1) \times U(1)$ Chern-Simons theories [38, 39, 21]. These theories, denoted by $(Z_N)_k$, are labelled by their Chern-Simons level $k \sim k + 2N$. They include all possible bosonic $\mathbb{Z}_N$ gauge theories classified by $H^3(\mathbb{Z}_N, U(1)) = \mathbb{Z}_N$ and some fermionic gauge theories. Their Lagrangian is

$$\frac{k}{4\pi} \widehat{a} d\widehat{a} + \frac{N}{2\pi} \widehat{a} d\widehat{b} . \tag{5.1}$$

Integrating out the gauge field $\widehat{b}$ constrains $\widehat{a}$ to be a $\mathbb{Z}_N$ gauge field $\oint \widehat{a} \in \frac{2\pi}{N} \mathbb{Z}$.

We define the electric and magnetic line operators as $U = \exp(i \oint \widehat{a})$ and $V = \exp(i \oint \widehat{b})$ respectively. They obey the relation

$$U^N = 1, \qquad U^k V^N = \psi^k , \tag{5.2}$$

where $\psi$ is the transparent fermion line. For odd $k$, the theory contains $\psi$, and hence it is a fermionic theory that depends on spin structure. For simplicity, we will restrict to bosonic $\mathbb{Z}_N$ gauge theories, *i.e* theories with even $k$ below.

The line operators form an Abelian group. The group can be understood as the quotient of $\mathbb{Z} \times \mathbb{Z}$ by the group generated by the columns of the following matrix

$$\begin{pmatrix} N & k \\ 0 & N \end{pmatrix} . \tag{5.3}$$

The matrix can be put into Smith normal form

$$\begin{pmatrix} L & 0 \\ 0 & N^2/L \end{pmatrix} \tag{5.4}$$

with $L = \gcd(k, N)$ by multiplying $SL(2, \mathbb{Z})$ matrices from the left and the right. The resulting quotient group is invariant under the transformation. Hence for even $k$ the line operators generate a $\mathcal{A} = \mathbb{Z}_L \times \mathbb{Z}_{N^2/L}$ one-form symmetry (for odd $k$ the one-form symmetry will be modified by the additional $\mathbb{Z}_2$ symmetry generated by $\psi$). We emphasize that the one-form symmetry $\mathcal{A}$ always has a $\mathbb{Z}_N$ subgroup generated by $U$. This $\mathbb{Z}_N$ subgroup will be important in the later discussions.[17]

We can couple the one-form symmetry $\mathcal{A}$ to background gauge fields as follows. Let $B_2^e$ be a $\mathbb{Z}_N$ two-cocycle and $B_2^m$ be a $\mathbb{Z}_N$ two-cochain. The background $B_2^e$ couples to the theory by modifying the quantization of $a$

$$\oint \frac{d\widehat{a}}{2\pi} = \frac{1}{N} \oint B_2^e \mod \mathbb{Z} , \tag{5.5}$$

while the background $B_2^m$ couples to the $\mathbb{Z}_N$ one-form symmetry generated $U = \exp(i \oint \widehat{a})$.

In continuous notation, we can embed the discrete $\mathbb{Z}_N$ background gauge fields into $U(1)$ gauge fields $\widehat{B}_2^e$ and $\widehat{B}_2^m$ with

$$\oint \widehat{B}_2^e = \frac{2\pi}{N} \oint B_2^e \mod 2\pi\mathbb{Z}, \quad \oint \widehat{B}_2^m = \frac{2\pi}{N} \oint B_2^m \mod 2\pi\mathbb{Z} . \tag{5.6}$$

Then the background gauge fields can couple to the theory by adding the following term to the Lagrangian (5.1)

$$\frac{N}{2\pi}\widehat{a}\widehat{B}_2^m + \frac{N}{2\pi}\widehat{b}\widehat{B}_2^e . \tag{5.7}$$

Integrating out $\widehat{b}$ imposes the constraint (5.5) which implies that the gauge field $\widehat{a}$ is no

---

[17]The theory can also have non-trivial zero-form symmetries that permutes the line operators (see *e.g.* [40]).

longer properly quantized. Hence the remaining action develops a bulk dependence

$$
\int \left( \frac{k}{4\pi} d(\widehat{a}d\widehat{a}) + \frac{N}{2\pi} d(\widehat{a}\widehat{B}_2^m) \right)
$$
$$
= \frac{N}{2\pi} \int \widehat{a} \left( \frac{k}{N} d\widehat{B}_2^e - d\widehat{B}_2^m \right) + \int \left( \frac{N}{2\pi} \widehat{B}_2^e \widehat{B}_2^m - \frac{k}{4\pi} \widehat{B}_2^e \widehat{B}_2^e \right) \mod 2\pi\mathbb{Z} . \tag{5.8}
$$

The first term involves dynamical gauge fields so it has to be removed. This can be achieved by imposing the following constraint on the backgrounds

$$
N \frac{d\widehat{B}_2^m}{2\pi} = k \frac{d\widehat{B}_2^e}{2\pi} . \tag{5.9}
$$

The second term of (5.8) depends only on the background fields. It represents an 't Hooft anomaly,

$$
\int \left( \frac{N}{2\pi} B_2^e B_2^m - \frac{k}{4\pi} B_2^e B_2^e \right) . \tag{5.10}
$$

The anomaly is also given by the spin of the symmetry line operators [13].

The above calculation is repeated in discrete notation in appendix B. In the discrete notation, the backgrounds obay

$$
\delta B_2^m = k \mathrm{Bock}(B_2^e) , \tag{5.11}
$$

where Bock is the Bockstein homomorphism for the exact sequence $1 \to \mathbb{Z}_N \to \mathbb{Z}_{N^2} \to \mathbb{Z}_N \to 1$. The anomaly is

$$
\frac{2\pi}{N} \int B_2^e B_2^m - \frac{2\pi k}{2N^2} \int \mathcal{P}(B_2^e) . \tag{5.12}
$$

## 5.2 Fermionic $\mathbb{Z}_2$ one-form gauge theory

Fermionic $\mathbb{Z}_2$ gauge theory in $3d$ can be constructed by gauging the $\mathbb{Z}_2$ symmetry in the $3d$ SPT phase with unitary $\mathbb{Z}_2$ symmetry, the later admits $\mathbb{Z}_8$ classification [41–44, 17, 45] from $\Omega^3_{\mathrm{Spin}}(B\mathbb{Z}_2) = \mathbb{Z}_8$. The Abelian Chern-Simons theory construction discussed above only accounts for four of them. The other four fermionic gauge theories have non-Abelian anyons. All these $\mathbb{Z}_2$ gauge theories, denoted by $(\mathbb{Z}_2)_L$, can be described by (see Appendix B of [3])

$$
(\mathbb{Z}_2)_L \quad \longleftrightarrow \quad Spin(L)_{-1} \times SO(L)_1 , \tag{5.13}
$$

where $L \sim L + 8$. The $\mathbb{Z}_2$ gauge theory $(Z_2)_k$, that has an Abelian Chern-Simons theory construction (5.1), is mapped to $(\mathbb{Z}_2)_{2k}$ by tensoring with an almost trivial theory $\{1, \psi\}$ with $\psi$ the transparent fermion line.

The $Spin(L)_1$ TQFT was studied in [46] (also see *e.g* Appendix C of [47] for a review). For odd $L$, the $Spin(L)_1$ theory has three lines: the identity line 1 with spin 0, the line $\epsilon$ in vector representation with spin $\frac{1}{2}$ and the line $\sigma$ in spinor representation with spin $-\frac{L}{16}$. They obey the Ising fusion rule:

$$\epsilon \times \epsilon = 1, \quad \sigma \times \sigma = 1 + \epsilon, \quad \sigma \times \epsilon = \sigma . \tag{5.14}$$

The product of $\epsilon$ and $\psi$ is mapped to the Wilson line of $(\mathbb{Z}_2)_L$, which generates a $\mathbb{Z}_2$ one-form symmetry. This $\mathbb{Z}_2$ one-form symmetry is crucial in the later discussion. The only charged line under this symmetry is $\sigma$.

To conclude, we summarize the fusion rule and the one-form symmetry of $(\mathbb{Z}_2)_L$ gauge theory (omitting the transparent fermion line $\psi$ with $\psi^2 = 1$ from the $SO(L)_1$ factor in (5.13))

- $L = 0$ mod 4: the theory has topological lines that obey $\mathbb{Z}_2 \times \mathbb{Z}_2$ fusion rule. They generate a $\mathbb{Z}_2 \times \mathbb{Z}_2$ one-form symmetry.

- $L = 2$ mod 4: the theory has topological lines that obey $\mathbb{Z}_4$ fusion rules. They generate a $\mathbb{Z}_4$ one-form symmetry.

- $L = 1$ mod 2: When $L = 1, 7$ mod 8, the theory has topological lines that form a $\mathbb{Z}_2$ Tambara-Yamagami category TY$_+$. When $L = 3, 5$ mod 8, the theory has topological lines that form another $\mathbb{Z}_2$ Tambara-Yamagami category TY$_-$. The two Tambara-Yamagami categories TY$_\pm$ have the same Ising fusion rule, but different $F$-symbols [48]. Among these topological lines, $1, \epsilon$ generate $\mathbb{Z}_2$ one-form symmetry, while $\sigma$ is a non-invertible topological line.

## 5.3 Couple QFT to bosonic one-form gauge theory

Consider a 3$d$ theory with a non-anomalous $\mathbb{Z}_N$ 0-form symmetry. Gauging the symmetry with or with adding an SPT phase leads to different theories. Denote the resulting theory with a Chern-Simons level $k$ for the $\mathbb{Z}_N$ gauge field by $\mathcal{T}^k$. Below we will restrict to bosonic SPTs i.e. theories with even $k$.

The theory $\mathcal{T}^k$ has a $\mathbb{Z}_N$ one-form symmetry generated by $U = \exp(\frac{2\pi i}{N} \oint a)$ where $a$ is the dynamical $\mathbb{Z}_N$ gauge field. The $\mathbb{Z}_N$ one-form symmetry can be understood as the emergent symmetry dual to the gauged $\mathbb{Z}_N$ zero-form symmetry. All the gauged theories are related by

$$\mathcal{T}^{p+k} \quad \longleftrightarrow \quad \frac{\mathcal{T}^p \times (Z_N)_k}{\mathbb{Z}_N^{(1)}} , \tag{5.15}$$

where the quotient means gauging the diagonal $\mathbb{Z}_N$ one-form symmetry that identifies the $\mathbb{Z}_N$ gauge fields in $\mathcal{T}^p$ and $(Z_N)_k$.

The equation (5.15) is compatible with the addition of discrete theta angle. If we apply (5.15) again with $k$ replaced by $k'$,

$$
\mathcal{T}^{k+k'} \quad \longleftrightarrow \quad \frac{\mathcal{T}^k \times (Z_N)_{k'}}{\mathbb{Z}_N^{(1)}} \quad \longleftrightarrow \quad \frac{\mathcal{T}^0 \times (Z_N)_k \times (Z_N)_{k'}}{\mathbb{Z}_N^{(1)} \times \mathbb{Z}_N^{(1)}} \quad \longleftrightarrow \quad \frac{\mathcal{T}^0 \times (Z_N)_{k+k'}}{\mathbb{Z}_N^{(1)}} \ ,
$$

(5.16)

where in the last duality we reparametrized the $\mathbb{Z}_N^{(1)} \times \mathbb{Z}_N^{(1)}$ quotient such that one of them acts only on $(Z_N)_k \times (Z_N)_{k'}$ and identifies their $\mathbb{Z}_N$ gauge fields to give $(Z_N)_{k+k'}$.

Let us compare the symmetries in $\mathcal{T}^0$ and $\mathcal{T}^k$. The two theories in general may not have the same symmetry. This arises if the symmetry in $\mathcal{T}^0$ has a mixed anomaly with the dual $\mathbb{Z}_N$ one-form symmetry.

Suppose the theory $\mathcal{T}^0$ has a one-form symmetry $\mathcal{A}$, which is a group extension

$$
1 \to \mathbb{Z}_N \to \mathcal{A} \to \mathbb{Z}_r \to 1 \ ,
$$

(5.17)

specified by a $\mathbb{Z}_N$ element $k'$. In terms of the symmetry generators, this means

$$
U^N = 1, \quad V^r = U^{k'} \ ,
$$

(5.18)

where $U$ and $V$ are the generator of the $\mathbb{Z}_N$ and $\mathbb{Z}_r$ one-form symmetry. The $\mathbb{Z}_N$ subgroup one-form symmetry should be identified with the $\mathbb{Z}_N$ one-form symmetry generated by the Wilson line $U = \exp(\frac{2\pi i}{N} \oint a)$. We will refer to this symmetry as the $\mathbb{Z}_N$ magnetic one-form symmetry. The background gauge field for the one-form symmetry $\mathcal{A}$ can be described by a $\mathbb{Z}_r$ cocycle $B_2$ and a $\mathbb{Z}_N$ cochain $B_2^m$ with the constraint

$$
\delta B_2^m = k' \text{Bock}(B_2) \ ,
$$

(5.19)

where Bock is the Bockstein homomorphism for the short exact sequence $1 \to \mathbb{Z}_r \to \mathbb{Z}_{Nr} \to \mathbb{Z}_r \to 1$.

We further assume that the symmetry generator $U$ and $V$ has non-trivial mutual braiding, which implies an 't Hooft anomaly of the one-form symmetry $\mathcal{A}$ [13]. The 't Hooft anomaly becomes trivial when it is restricted to the $\mathbb{Z}_N$ subgroup one-form symmetry (gauging the $\mathbb{Z}_N$ one-form symmetry recovers the original theory). This implies that $V^r = U^{k'}$ has trivial braiding with $U$ so the braiding between $U$ and $V$ can only be $\exp(2\pi i q/\gcd(N,r))$ for some integer $q$. This leads to the mixed anomaly [13]

$$
\frac{2\pi q}{\gcd(N,r)} \int_{4d} B_2 B_2^m \ ,
$$

(5.20)

which can be accompanied by an anomaly $\omega(B_2)$ that depends only on $B_2$.

Let us now discuss the symmetry of the theory $\mathcal{T}^k$. The theory $\mathcal{T}^k$ is constructed by gauging the diagonal $\mathbb{Z}_N$ one-form symmetry in the theory $\mathcal{T}^0 \times (Z_N)_k$. The gauging sets the gauge fields for the magnetic one-form symmetries to be $b_2^m$ in $\mathcal{T}^0$ and $b_2^m + B_2^m$ in $(Z_N)_k$. Here $b_2^m$ is a dynamical gauge field, and $B_2^m$ is the background gauge field for the residue magnetic one-form symmetry. The theory has the bulk dependence

$$\frac{2\pi q}{\gcd(N, r)} \int_{4d} B_2 b_2^m + \frac{2\pi}{N} \int_{4d} B_2^e (b_2^m + B_2^m) - \frac{2\pi k}{2N^2} \int_{4d} \mathcal{P}(B_2^e) + \omega(B_2) . \tag{5.21}$$

To cancel the gauge-global anomaly $i.e$ the bulk terms that depend on $b_2^m$, the background must satisfy

$$B_2^e = -\frac{Nq}{\gcd(N, r)} B_2 . \tag{5.22}$$

The gauge fields for the magnetic one-form symmetries further obey the constraints (5.11), (5.19)

$$\delta(b_2^m + B_2^m) = k'\mathrm{Bock}(B_2), \quad \delta b_2^m = k\mathrm{Bock}(B_2^e) , \tag{5.23}$$

Together with (5.22) we find the relation

$$\delta B_2^m = k'\mathrm{Bock}(B_2) - k\mathrm{Bock}(B_2^e) = \left(k' + k\frac{qr}{\gcd(r, N)}\right) \mathrm{Bock}(B_2) . \tag{5.24}$$

It implies that the one-form symmetry of $\mathcal{T}^k$ is

$$\mathbb{Z}_J \times \mathbb{Z}_{Nr/J}, \quad J = \gcd\left(k' + \frac{qr}{\gcd(N, r)}k, N, r\right) . \tag{5.25}$$

The symmetry has an anomaly obtained by substituting (5.22) back to (5.21)

$$-\frac{2\pi q}{\gcd(N, r)} \int_{4d} B_2 B_2^m - \frac{2\pi k q^2}{2\gcd(N, r)^2} \int_{4d} \mathcal{P}(B_2) + \omega(B_2) . \tag{5.26}$$

As a check, consider gauging a $\mathbb{Z}_N$ zero-form symmetry in an empty theory with an additional Chern-Simons term for the $\mathbb{Z}_N$ gauge field. This leads to a family of theories $\mathcal{T}^p = (Z_N)_p$. From the symmetry extension (5.11) and 't Hooft anomaly (5.12), we identified the theory $\mathcal{T}^p$ as a special case of the discusssion above with $k' = p$, $r = N$, $q = 1$. The above analysis then implies that the theory $\mathcal{T}^{p+k}$ has the symmetry

$$\mathbb{Z}_J \times \mathbb{Z}_{N^2/J}, \quad J = \gcd\left(k + p, N\right) , \tag{5.27}$$

which agrees with the one-form symmetry of the theory $\mathcal{T}^{p+k} = (Z_N)_{p+k}$.

### 5.3.1 Example: gauging $\mathbb{Z}_N$ zero-form symmetry in $\mathbb{Z}_M$ gauge theory

A $3d$ $\mathbb{Z}_M$ gauge theory with the Lagrangian

$$\frac{M}{2\pi}\widehat{u}d\widehat{v} \ , \tag{5.28}$$

can be coupled to a $\mathbb{Z}_N$ one-form gauge field $\widehat{a}$ through

$$\frac{1}{2\pi}\widehat{a}(Nd\widehat{b} - d\widehat{v}) \ , \tag{5.29}$$

where $\widehat{b}$ is a dynamical gauge field that constrains $\widehat{a}$ to be a $\mathbb{Z}_N$ gauge field $\oint \widehat{a} \in \frac{2\pi}{N}\mathbb{Z}$.

Promoting $\widehat{a}$ to a dynamical gauge field gauges a $\mathbb{Z}_N$ zero-form symmetry. The dynamical gauge field $\widehat{a}$ then becomes a Lagrange multiplier that forces $d\widehat{v} = Md\widehat{b}$ which gives a $\mathbb{Z}_{NM}$ gauge theory

$$\frac{NM}{2\pi}\widehat{u}d\widehat{b} \ . \tag{5.30}$$

The theory has a $\mathbb{Z}_N$ one-form symmetry generated by $\exp(i\oint \widehat{a}) = \exp(iM\oint \widehat{u})$ (which is the $\mathbb{Z}_N$ subgroup of a larger $\mathbb{Z}_{NM}$ one-form symmetry generated by $\exp(i\oint \widehat{u})$), and a $\mathbb{Z}_{NM}$ one-form symmetry generated by $\exp(i\oint \widehat{b})$. These two symmetries have a mixed anomaly due to the non-trivial braiding phase $\exp(2\pi i/N)$ between their generators. Denote their background gauge field by $B_2^m$ and $B_2$ respectively. The anomaly is characterized by [13]

$$\frac{2\pi}{N}\int_{4d} B_2^m B_2 \ . \tag{5.31}$$

Comparing with the discussion above, we identify $r = MN$ and $q = 1$, $k' = 0$.

We can add a Chern-Simons term with level $k$ for the $\mathbb{Z}_N$ gauge field. Applying the analysis above, we find that the resulting theory has the following one-form symmetry

$$\mathbb{Z}_J \times \mathbb{Z}_{N^2M/J}, \quad J = \gcd(kM, N) \ . \tag{5.32}$$

The one-form symmetry has an 't Hooft anomaly

$$\frac{2\pi}{N}\int_{4d} B_2 B_2^m - \frac{2\pi k}{2N^2}\int_{4d} \mathcal{P}(B_2) \ . \tag{5.33}$$

As a consistent check, we can examine the one-form symmetry of the resulting theory directly. It has Lagrangian

$$\frac{M}{2\pi}\widehat{u}d\widehat{v} + \frac{1}{2\pi}\widehat{a}(Nd\widehat{b} - d\widehat{v}) + \frac{k}{4\pi}\widehat{a}d\widehat{a} \ . \tag{5.34}$$

Integrating out the gauge field $\widehat{v}$ simplifies the theory to

$$\frac{NM}{2\pi}\widehat{u}d\widehat{b} + \frac{kM^2}{4\pi}\widehat{u}d\widehat{u} \; . \tag{5.35}$$

The theory has a $\mathbb{Z}_J \times \mathbb{Z}_{N^2 M/J}$ subgroup symmetry generated by $\exp(iM \oint \widehat{u})$ and $\exp(i \oint \widehat{b})$ whose spins are consistent with the 't Hooft anomaly (5.33).

## 5.4 Couple QFT to fermionic $\mathbb{Z}_2$ gauge theory

Consider gauging a non-anomalous $\mathbb{Z}_2$ zero-form symmetry in a $3d$ system. We can add an additional fermionic SPT phase for the $\mathbb{Z}_2$ symmetry classified by $\Omega^3_{\text{Spin}}(B\mathbb{Z}_2) = \mathbb{Z}_8$. Denote the theory with discrete theta angle $k$ by $\mathcal{T}^k$. All these theories are related by

$$\mathcal{T}^k \quad \longleftrightarrow \quad \frac{\mathcal{T}^0 \times (\mathbb{Z}_2)_k}{\mathbb{Z}_2^{(1)}} \quad \longleftrightarrow \quad \frac{\mathcal{T}^0 \times Spin(k)_{-1} \times SO(k)_1}{\mathbb{Z}_2^{(1)}} \; , \tag{5.36}$$

where the quotient means gauging the diagonal $\mathbb{Z}_2$ one-form symmetry generated by the product of $\exp(i\pi \oint a)$ in $\mathcal{T}^0$, $\epsilon$ in $Spin(k)_{-1}$ and $\psi$ in $SO(k)_1$.

### 5.4.1 Non-invertible topological lines

As discussed in the previous subsection, the theory $\mathcal{T}^0$ and $\mathcal{T}^k$ with even $k$ can have different symmetry. This occurs when $\mathcal{T}^0$ has another $\mathbb{Z}_2$ one-form symmetry whose generator carries charge 1 under the emergent $\mathbb{Z}_2$ one-form symmetry generated by $\exp(i\pi \oint a)$. In the theory $\mathcal{T}^k$, the generator of the $\mathbb{Z}_2$ one-form symmetry is paired with the lines in the spinor representation of $Spin(k)_{-1}$ that are also odd under the $\mathbb{Z}_2$ one-form symmetry due to the gauging in (5.36). This leads to the following fusion category in the theory $\mathcal{T}^k$:

- $k = 0 \mod 4$: the theory has topological lines that obey $\mathbb{Z}_2 \times \mathbb{Z}_2$ fusion rule. They generate a $\mathbb{Z}_2 \times \mathbb{Z}_2$ one-form symmetry.

- $k = 2 \mod 4$: the theory has topological lines that obey $\mathbb{Z}_4$ fusion rules. They generate a $\mathbb{Z}_4$ one-form symmetry.

- $k = 1 \mod 2$: When $k = 1, 7 \mod 8$, the theory has topological lines that form a $\mathbb{Z}_2$ Tambara-Yamagami category $\text{TY}_+$. When $k = 3, 5 \mod 8$, the theory has topological lines that form another $\mathbb{Z}_2$ Tambara-Yamagami category $\text{TY}_-$. The two Tambara-Yamagami categories $\text{TY}_\pm$ have the same Ising fusion rule, but different $F$-symbols [48]. Among these topological lines, $1, \epsilon$ generate $\mathbb{Z}_2$ one-form symmetry, while $\sigma$ is a non-invertible topological line.

We remark that the above discussion can be compared with the discussion in Section 6 of [49] for $2d$ fermionic CFT equipped with an anomalous $\mathbb{Z}_2$ symmetry (classified by $\mathbb{Z}_8$ [41–44,17,45]) that has a mixed anomaly with the total fermion parity. It was shown that after gauging the total fermion parity the $\mathbb{Z}_2$ symmetry gets extended, with the symmetry line defect turned into one of the line in the above fusion categories with $k$ identified with the $\mathbb{Z}_8$ anomaly coefficient in $2d$. This can be understood from gauging the total fermion parity in a $2d/3d$ boundary/bulk system, with the $3d$ system given by product of a spin theory and the fermionic SPT phase for the $\mathbb{Z}_2$ symmetry. The line in the $3d$ can move to the $2d$ boundary.[18]

# 6    $3d$ gauge theories with discrete theta angles

In this section we discuss concrete examples of gauging a non-anomalous $\mathbb{Z}_N$ zero-form symmetry in $3d$ theories with an additional SPT phase that becomes a discrete theta angle. We also discuss the symmetry in $O(N)$ Chern-Simons theory with discrete theta angle denoted by $O(N)^1$ in the notation of [3].

## 6.1    Gauging $\mathbb{Z}_N \subset G$ subgroup zero-form symmetry

We start with a system in $3d$ with a 0-form symmetry $\widetilde{G}$ which is an extension of $G$ by $\mathbb{Z}_N$

$$1 \to \mathbb{Z}_N \to \widetilde{G} \to G \to 1 \ . \tag{6.1}$$

We assume the $\mathbb{Z}_N$ subgroup symmetry is non-anomalous and there is no mixed anomaly between $\mathbb{Z}_N$ and $\widetilde{G}$. Then we gauge the $\mathbb{Z}_N$ subgroup symmetry with an additional SPT phase given by a level $k$ Chern-Simons term. What's the symmetry of the new system?

For $k = 0$ the new system has an emergent $\mathbb{Z}_N$ dual one-form symmetry generated by the $\mathbb{Z}_N$ Wilson line. The extension $\widetilde{G}$ implies that this emergent one-form symmetry has a mixed anomaly with the remaining $G$ 0-form symmetry. To see this, we can turn on background gauge field $B_1$ for $G$, and and background gauge field $B_2$ for the $\mathbb{Z}_N$ one-form symmetry. Denote the (dynamical) $\mathbb{Z}_N$ one-form gauge field by $a$, then the symmetry extension $\widetilde{G}$ implies that

$$\delta a = B_1^* \eta_2 \ , \tag{6.2}$$

where $\eta_2 \in H^2(G, \mathbb{Z}_N)$ describes the group extension $\widetilde{G}$. The background $B_2$ couples as

$$\frac{2\pi}{N} \int a B_2 \ . \tag{6.3}$$

---

[18]We thank Shu-Heng Shao for pointing this out to us.

Thus the coupling has a mixed anomaly described by the bulk term

$$\frac{2\pi}{N} \int B_1^* \eta_2 \, B_2 \; . \tag{6.4}$$

Now, let us consider theories with nonzero Chern-Simons term $k$. Comparing (6.2), (6.3) with (5.5),(5.7), we identify $B_2^e = B_1^* \eta_2$ and $B_2^m = B_2$. Thus these backgrounds satisfy

$$\delta B_2 = k B_1^* \text{Bock}(\eta_2) \; , \tag{6.5}$$

which represents a two-group symmetry that combines the $\mathbb{Z}_N$ one-form symmetry and $G$ 0-form symmetry, with Postnikov class

$$\Theta = k \text{Bock}(\eta_2) \; . \tag{6.6}$$

The 't Hooft anomaly for the two-group symmetry is described by the bulk term

$$\frac{2\pi}{N} \int B_1^* \eta_2 \, B_2 - \frac{2\pi k}{2N^2} \int B_1^* \mathcal{P}(\eta_2) \; . \tag{6.7}$$

### 6.1.1 Example: $\mathbb{Z}_2$ gauge theory with two complex scalars

As an example, we consider gauging a $\mathbb{Z}_2$ zero-form symmetry (without Dijkgraaf-Witten action) in a theory with two complex scalars that are $\mathbb{Z}_2$ odd. The resulting theory $\mathcal{T}^0$ is a $\mathbb{Z}_2$ gauge theory with two charged complex scalars.

The theory has a magnetic $\mathbb{Z}_2$ one-form symmetry generated by the $\mathbb{Z}_2$ Wilson line, whose background is denoted by $B_2$. It also has an $SO(3)$ flavor symmetry that transforms the two complex scalars, whose background is denoted by $B_1$. The transformation that flips the signs of both scalars is identified with a gauge rotation. Thus if we turn on $SO(3)$ background gauge field that is not an $SU(2)$ background gauge field, whose obstruction is described by a non-trivial $w_2^f$, the $\mathbb{Z}_2$ gauge bundle will be twisted:

$$\delta a = B_1^* w_2^f \; . \tag{6.8}$$

Now, let us introduce a discrete theta angle for the $\mathbb{Z}_2$ bundle as described by Chern-Simons level $k$. We find that the background for the magnetic one-form symmetry (generated by the $\mathbb{Z}_2$ Wilson line) now satisfies

$$\delta B_2 = k B_1^* \text{Bock}(w_2^f) \; . \tag{6.9}$$

The relation describes a 2-group symmetry that combines the $\mathbb{Z}_2$ magnetic one-form symmetry and the $SO(3)$ flavor symmetry, with Postnikov class $\Theta = k \text{Bock}(w_2^f)$ that depends

on the discrete theta angle $k$. For even $k$, the Postnikov class is trivial so the symmetries do not combine into a two-group symmetry. The symmetry has an 't Hooft anomaly determined by (6.7):

$$\pi \int B_1^* w_2^f B_2 - \frac{k\pi}{4} \int B_1^* \mathcal{P}(w_2^f) \ . \tag{6.10}$$

If the scalars are massive with equal mass, the theory flows to a pure $\mathbb{Z}_2$ gauge theory $(Z_2)_k$ in the infrared. The infrared theory has an accidental electric one-form symmetry. To match the ultraviolet symmetry and anomaly, the $SO(3)$ gauge field $B_1$ couples to the infrared theory by a symmetry enrichment

$$B_2^e = B_1^* w_2^f \ , \tag{6.11}$$

using the background gauge field $B_2^e$ for the accidental electric one-form symmetry.

## 6.2 $O(N)$ Chern-Simons theory with discrete theta angle

Here we present an example with discrete theta angle associated to mixed topological terms that arise from gauging a $\mathbb{Z}_2$ one-form and a $\mathbb{Z}_2$ 0-form symmetry.

We start with a $Spin(N)_K$ Chern-Simons theory, and gauge the $\mathbb{Z}_2$ zero-form charge conjugate symmetry and the $\mathbb{Z}_2$ one-form symmetry that does not transform the Wilson lines in vector representation. The resulting theory is a $O(N)_K$ Chern-Simons theory. We can add to the theory a discrete theta angle

$$p \int w_1 \cup w_2^{(1)} \ , \tag{6.12}$$

where $p$ is a $\mathbb{Z}_2$ coefficient. The characteristic classes $w_1, w_2^{(1)}$ are defined in section 4. They are controlled by the dynamical gauge fields for the zero-form charge conjugation symmetry and the one-form symmetry, respectively. We will focus on theories with even $N, K$.

The theory $O(N)_K$ with $p = 0$ has one-form symmetry [3]

$$\mathcal{A} = \begin{cases} \mathbb{Z}_2 \times \mathbb{Z}_2 & K = 0 \bmod 4 \\ \mathbb{Z}_4 & K = 2 \bmod 4 \end{cases} \ . \tag{6.13}$$

The $\mathbb{Z}_2$ subgroup of the one-form symmetry is generated by the symmetry line operator $\exp(i\pi \oint w_1)$, with background denoted by $B_2^{(3)}$. There is also center one-form symmetry, with background denoted by $B_2^{(2)}$. The one-form symmetry is the extension of these two symmetries. It is reflected in the constraint of the backgrounds

$$\delta B_2^{(3)} = \frac{K}{2} \text{Bock}(B_2^{(2)}) \ , \tag{6.14}$$

where Bock is the Bockstein homomorphism for the exact sequence $1 \to \mathbb{Z}_2 \to \mathbb{Z}_4 \to \mathbb{Z}_2 \to 1$. The symmetry extension implies that if the constraint were not satisfied, rather $\delta B_3 = 0$, the theory has a mixed gauge-global anomaly given by the bulk term

$$\frac{K}{2}\pi \int w_1 \cup \text{Bock}(B_2^{(2)}) . \tag{6.15}$$

What's the symmetry in the theory with $p = 1$? As explained in section 4, the background $B_2^{(2)}$ modifies the cocycle condition of $w_2^{(1)}$ such that

$$\delta w_2^{(1)} = \frac{N}{2}\text{Bock}(B_2^{(2)}) + B_2^{(2)}w_1 . \tag{6.16}$$

Thus the discrete theta angle (6.12) with $p = 1$ has the bulk dependence

$$\pi \int \delta \left( w_1 \cup w_2^{(2)} \right) = \pi \int w_1 \cup \left( \frac{N}{2}\text{Bock}(B_2^{(2)}) + B_2^{(2)}w_1 \right)$$
$$= \frac{N+2}{2}\pi \int w_1 \cup \text{Bock}(B_2^{(2)}) , \tag{6.17}$$

where we used the property $w_1 \cup w_1 = \text{Bock}(w_1)$ and $\pi \int \text{Bock}(B_2^{(2)}w_1)$ is trivial on orientable manifolds. Thus in order to cancel the gauge-global anomaly, the background field $B_2^{(3)}$ must obey the new condition

$$\delta B_2^{(3)} = \frac{K+N+2}{2}\text{Bock}(B_2^{(2)}) , \tag{6.18}$$

which implies that the one-form symmetry in the theory with $p = 1$ is

$$\mathcal{A} = \begin{cases} \mathbb{Z}_2 \times \mathbb{Z}_2 & K+N+2 = 0 \bmod 4 \\ \mathbb{Z}_4 & K+N+2 = 2 \bmod 4 \end{cases} \tag{6.19}$$

in agreement with [3] and consistent with the level-rank dualities [3].

This example can also be understood as follows. Consider the $3d$ TQFT (C.1) with $N = 2$, $q = 0$ and $p = 1$. The theory can be expressed as

$$\mathcal{T}^1 \quad \longleftrightarrow \quad \frac{\mathcal{T}^0 \times \text{TQFT}}{\mathbb{Z}_2^{(0)} \times \mathbb{Z}_2^{(1)}} , \tag{6.20}$$

where the quotient $\mathbb{Z}_2^{(0)}$ means gauging the diagonal 0-form symmetry that identifies $w_2^{(1)}$ with $b$ in the TQFT, and $\mathbb{Z}_2^{(1)}$ means gauging the diagonal one-form symmetry that identifies $w_1$ with $a$ in the TQFT. The condition (6.16) implies that the TQFT is coupled to the gauge field $Y_3 = \text{Bock}(B_2^{(2)}) + B_2^{(2)}a$. Then a similar computation in the TQFT shows that the symmetry is deformed as we discussed above.

# 7 $2d$ $\mathbb{Z}_2$ one-form gauge theory

$2d$ $\mathbb{Z}_2$ fermionic gauge theory can be constructed by gauging $\mathbb{Z}_2$ symmetry in $2d$ fermionic SPT phase with unitary $\mathbb{Z}_2$ symmetry (in addition to the fermion parity). The latter admits $\mathbb{Z}_2 \times \mathbb{Z}_2$ classification [17], while one of the $\mathbb{Z}_2$ is generated by the fermionic SPT phase without the $\mathbb{Z}_2$ symmetry [18, 19, 17] (given by the Arf invariant [15, 16]). Thus there are two fermionic $\mathbb{Z}_2$ gauge theories in $2d$, labelled by discrete theta angle $p = 0, 1$.

The action for the $\mathbb{Z}_2$ gauge theory can be constructed from the Arf invariant $\mathrm{Arf}(\rho)$, which is a $\mathbb{Z}_2$ function of the spin structure $\rho$. The spin structure $\rho$ is a $\mathbb{Z}_2$ one-cochain that trivializes the second Siefel-Whitney class of the tangent bundle $w_2(TM) = \delta\rho$. Denote the $\mathbb{Z}_2$ gauge field by $a$ which is a $\mathbb{Z}_2$ one-cocycle. Define

$$q(a) = \mathrm{Arf}(a + \rho) - \mathrm{Arf}(\rho) . \tag{7.1}$$

The action of the $\mathbb{Z}_2$ gauge theory with gauge field $a$ is

$$p\pi \, q(a), \quad p = 0, 1 . \tag{7.2}$$

We remark that $q$ is the quadratic refinement of the cup product [15, 16]: for any $\mathbb{Z}_2$ one-cocycles $a, b$,

$$q(a + b) = q(a) + q(b) + \int a \cup b \mod 2 . \tag{7.3}$$

Let us begin with $p = 0$. The theory has an emergent $\mathbb{Z}_2$ zero-form symmetry generated by the $\mathbb{Z}_2$ line operator $\exp(i\pi \oint a)$, whose background is denoted by $B_1$. The theory also has a $\mathbb{Z}_2$ one-form symmetry with background $B_2$, which modifies the cocycle condition for $a$ to be

$$\delta a = B_2 . \tag{7.4}$$

The coupling to $B_1$ is

$$\pi \int a \cup B_1 . \tag{7.5}$$

In the presence of $B_2$ the coupling depends on the bulk and it results a mixed anomaly between the zero-form and the one-form symmetries:

$$\pi \int \delta a \cup B_1 = \pi \int B_2 \cup B_1 . \tag{7.6}$$

Now let us discuss the case $p = 1$. In the presence of $B_2$, $a$ is no longer a $\mathbb{Z}_2$ cocycle and thus the action $\pi q(a)$ is not well-defined. However, the total action (7.2) and (7.5) can

be made well-defined if the backgrounds obey

$$B_2 = p\delta B_1 \ . \tag{7.7}$$

The total action together with an additional classical local counterterm $\pi q(B_1)$ combines into

$$\pi q(a + B_1) \ , \tag{7.8}$$

which is well-defined since $a + B_1$ is a $\mathbb{Z}_2$ cocycle. The constraint (7.7) implies that the background of the $\mathbb{Z}_2$ one-form symmetry is trivial for $p = 1$, and thus the one-form symmetry is explicitly broken in this case.

We remark that for $p = 1$ there is no 't Hooft anomaly for the above symmetries since the action (7.8) is well-defined in the presence of the background gauge fields. This is consistent with the fact that the theory with $p = 1$ is an invertible (spin-)TQFT [50] (it describes the Kitaev chain [18] as discussed in [17]), and thus all anomalies must be trivial by the 't Hooft anomaly matching condition.

## 7.1   Couple QFT to $\mathbb{Z}_2$ one-form gauge theory

Consider a $2d$ system with ordinary symmetry $\widetilde{G}$ that is the extension of $G$ by $\mathbb{Z}_2$,

$$1 \to \mathbb{Z}_2 \to \widetilde{G} \to G \to 1 \ . \tag{7.9}$$

The background gauge field for the $\widetilde{G}$ symmetry can be described by a $\mathbb{Z}_2$ cochain $a$ and background $B_1'$ for the $G$ symmetry, with the constraint

$$\delta a = (B_1')^* \eta_2 \ , \tag{7.10}$$

where $\eta_2 \in H^2(G, \mathbb{Z}_2)$ specifies the group extension $\widetilde{G}$.

In the following we assume the $\mathbb{Z}_2$ normal subgroup is non-anomalous and we will gauge this symmetry. We will also assume there is no mixed anomaly between the $\mathbb{Z}_2$ symmetry and $G$. We can include the discrete theta angle $p = 0, 1$ for the $\mathbb{Z}_2$ gauge field $a$ given by (7.2). The resulting system has a new $\mathbb{Z}_2$ 0-form symmetry generated by $\exp(i\pi \oint a)$ with background identified with $B_1$. The condition (7.10) identifies the background $B_2$ with

$$B_2 = (B_1')^* \eta_2 \ . \tag{7.11}$$

For $p = 0$, the resulting system has $\mathbb{Z}_2 \times G$ symmetry. From (7.6) and (7.11), the two

symmetries have a mixed anomaly

$$\pi \int (B'_1)^* \eta_2 \cup B_1 \ . \tag{7.12}$$

For $p = 1$, from (7.7) and (7.11) we find the backgrounds satisfy

$$B'^*_1 \eta_2 = \delta B_1 \ , \tag{7.13}$$

which describes the background for the symmetry extension $\widetilde{G}$. Thus the resulting system has $\widetilde{G}$ symmetry, in contrast to the symmetry $\mathbb{Z}_2 \times G$ for $p = 0$. Moreover, there is no mixed anomaly between the $\mathbb{Z}_2$ subgroup symmetry and $\widetilde{G}$.[19]

# Acknowledgement

We thank Yasunori Lee, Kantaro Ohmori, Nathan Seiberg, Shu-Heng Shao and Yuji Tachikawa for discussions. We thank Nathan Seiberg, Shu-Heng Shao, Yuji Tachikawa and Juven Wang for comments on a draft. The work of P.-S. H. is supported by the U.S. Department of Energy, Office of Science, Office of High Energy Physics, under Award Number DE-SC0011632, and by the Simons Foundation through the Simons Investigator Award. H.T.L. is supported by a Croucher Scholarship for Doctoral Study, a Centennial Fellowship from Princeton University and Physics Department of Princeton University.

# A    Steenrod square and higher cup product

In this appendix we summarize some facts about cochains and higher cup products. For more details, see *e.g.* [51] and the appendix A of [10].

We triangulate the spacetime manifold $M$ with simplices, where a $p$-simplex is the $p$-dimensional analogue of a triangle or tetrahedron (for $p = 0$ it is a point, $p = 1$ it is an edge, etc). The $p$-simplices can be described by its vertices $(i_0, i_1, \cdots i_p)$ where we pick an ordering $i_0 < i_1 < \cdots i_p$.

---

[19]We remark that when the $\mathbb{Z}_2$ symmetry being gauged is replaced by the fermion parity, the relation between the theories $p = 0, 1$ given by $\mathcal{T}^1 \longleftrightarrow (\mathcal{T}^0 \times \text{fermionic } \mathbb{Z}_2 \text{ gauge theory}) / \mathbb{Z}_2$ (the quotient denotes gauging a $\mathbb{Z}_2$ ordinary symmetry) reproduces the relation between (3.11) and (3.12) in [49] when the theories are treated as spin theories (*i.e.* tensoring the bosonic theories with an invertible spin TQFT given by the Arf invariant) using the identity (A.2) there (with $s$ in (A.2) identified with the $\mathbb{Z}_2$ gauge field $a$ in the gauge theory). Gauging the $\mathbb{Z}_2$ symmetry in $\mathcal{T}^0$ recovers the fermionic theory before summing over the spin structures, while gauging the diagonal $\mathbb{Z}_2$ symmetry in $(\mathcal{T}^0 \times \text{fermionic } \mathbb{Z}_2 \text{ gauge theory})$ produces another bosonic theory $\mathcal{T}^1$. Since the $\mathbb{Z}_2$ gauge theory is invertible, $(\mathcal{T}^0 \times \text{fermionic } \mathbb{Z}_2 \text{ gauge theory}) / \mathbb{Z}_2$ is also equivalent to gauging the $\mathbb{Z}_2$ symmetry in $\mathcal{T}^0$ with a local counterterm.

A simplicial $p$-cochain $f \in C^p(G, \mathcal{A})$ is a function on $p$-simplices taking values in an Abelian group $\mathcal{A}$ (we use additive notation for Abelian groups). For simplicity, we will take $\mathcal{A}$ to be a field (an Abelian group endowed with two products: addition and multiplication).

The coboundary operation on the cochains $\delta : C^p(M, \mathcal{A}) \to C^{p+1}(M, \mathcal{A})$ is defined by

$$(\delta f)(i_0, i_1, \cdots i_{p+1}) = \sum_{j=0}^{p+1} (-1)^j f(i_0, \cdots \widehat{i_j}, \cdots i_{p+1}) \tag{A.1}$$

where the hatted vertices are omitted. The coboundary operation is nilpotent $\delta^2 = 0$. When a cochain $x$ satisfies $\delta x = 0$, it is called a cocycle.

The cup product $\cup$ for $p$-cochain $f$ and $q$-cochain $g$ gives a $(p+q)$-cochain defined by

$$(f \cup g)(i_0 \cdots i_{p+q}) = f(i_0, \cdots i_p) g(i_p \cdots i_{p+q}) . \tag{A.2}$$

It is associative but not commutative. In this note we will omit writing the cup products. The higher cup product $f \cup_1 g$ is a $(p+q-1)$ cochain, defined by

$$(f \cup_1 g)(i_0 \cdots i_{p+q-1}) = \sum_{j=0}^{p-1} (-1)^{(p-j)(q+1)} f(i_0, \cdots i_j i_{j+q}, \cdots i_{p+q-1}) g(i_j, \cdots i_{j+q}) . \tag{A.3}$$

It is not associative and not commutative.

We have the following relations for a $p$ cochain $f$ and $q$ cochain $g$:

$$f \cup g = (-1)^{pq} g \cup f + (-1)^{p+q+1} [\delta(f \cup_1 g) - \delta f \cup_1 g - (-1)^p f \cup_1 \delta g]$$
$$\delta(f \cup g) = \delta f \cup g + (-1)^p f \cup \delta g$$
$$\delta(f \cup_1 g) = \delta f \cup_1 g + (-1)^p f \cup_1 \delta g + (-1)^{p+q+1} f \cup g + (-1)^{pq+p+q} g \cup f . \tag{A.4}$$

More generally,

$$f \cup_i g = (-1)^{pq-i} g \cup_i f + (-1)^{p+q-i-1} (\delta(f \cup_{i+1} g) - \delta f \cup_{i+1} g - (-1)^p f \cup_{i+1} \delta g)$$
$$\delta(f \cup_i g) = \delta f \cup_i g + (-1)^p f \cup_i \delta g + (-1)^{p+q-i} f \cup_{i-1} g + (-1)^{pq+p+q} g \cup_{i-1} f . \tag{A.5}$$

Similarly, if there is $G$ action on $\mathcal{A}$ given by $\rho : G \to \mathrm{Aut}(\mathcal{A})$, one can define a twisted coboundary operation that is nilpotent. Similarly the cup products $\cup, \cup_1$ can be modified. The rules (A.4) are still true (with $\delta$ meaning the twisted coboundary operation).

When the coefficient group is $\mathcal{A} = \mathbb{Z}_2$, there are additional operations in the cohomology called the Steenrod squares. For the purpose of this note we only need the operations $Sq^1$ and $Sq^2$. $Sq^i$ maps a $\mathbb{Z}_2$ $p$-cocycle to a $\mathbb{Z}_2$ $(p+i)$-cocycle. The definitions of $Sq^1$ and $Sq^2$

acting on $\mathbb{Z}_2$ $i$-cocycle $x_i$ are

$$Sq^1(x_1) = x_1 \cup x_1, \ Sq^1(x_2) = x_2 \cup_1 x_2, \ Sq^2(x_1) = 0, \ Sq^2(x_2) = x_2 \cup x_2, \ Sq^2(x_3) = x_3 \cup_1 x_3 \ . \tag{A.6}$$

In particular, $Sq^1$ acts on the cohomology the same way as the Bockstein homomorphism for the short exact sequence $1 \to \mathbb{Z}_2 \to \mathbb{Z}_4 \to \mathbb{Z}_2 \to 1$.

# B  Extension of symmetries in discrete notation

In this appendix, we repeat the calculation in the main text using discrete notation.

## B.1  $4d$ $\mathbb{Z}_N$ two-form gauge theory

In discrete notation, the $\mathbb{Z}_N$ two-form gauge field (denoted by $b_2$) is a $\mathbb{Z}_N$ two-cocycle. The action is

$$2\pi \frac{p}{2N} \int \mathcal{P}(b_2) \ . \tag{B.1}$$

where $pN$ is an even integer and $\mathcal{P}(b_2)$ is the generalized Pontryagin square operation. The operation is constructed as follows [14]

$$\mathcal{P}(b_2) = b_2 \cup b_2 \qquad \in H^4(M, \mathbb{Z}_N) \quad \text{for odd } N \ , \tag{B.2}$$

$$\mathcal{P}(b_2) = \widetilde{b}_2 \cup \widetilde{b}_2 - \delta \widetilde{b}_2 \cup_1 \widetilde{b}_2 \ \in H^4(M, \mathbb{Z}_{2N}) \ \text{ for even } N \ , \tag{B.3}$$

where $\widetilde{b}_2$ is an integer lift of $b_2$ and $M$ is the spacetime manifold.

The theory has a $\mathbb{Z}_N$ one-form and a $\mathbb{Z}_N$ two-form symmetry, with backgrounds $B_2, Y_3$. $Y_3$ modifies the cocycle condition for the two-form gauge field

$$\delta b_2 = Y_3 \ , \tag{B.4}$$

such that it becomes a two-cochain while $B_2$ couples to the theory as

$$\frac{2\pi}{N} \int b_2 B_2 \ . \tag{B.5}$$

### B.1.1  Even $N$

In the presence of the background field, the action (B.1) is no longer well-defined for even $N$. Suppose we change the lift $\widetilde{b}_2 \to \widetilde{b}_2 + N u_2$ for some integral 2-cochain $u_2$. The action

is shifted by

$$2\pi\frac{p}{2}\int\left(u_2\cup_1\delta\widetilde{b}_2-\delta\widetilde{b}_2\cup_1 u_2\right)\mod 2\pi\ ,\tag{B.6}$$

where we used $\delta u_2\cup_1 b_2=\delta(u_2\cup_1 b_2)-u_2\cup_1\delta b_2+u_2\cup b_2-b_2\cup u_2$. Since $b_2$ is no longer a two-cocycle, the shift does not vanish, instead, it gives

$$2\pi\frac{p}{2}\int Y_3\cup_2\delta u_2\mod 2\pi\ ,\tag{B.7}$$

where we used $Y_3\cup_1 u_2+u_2\cup_1 Y_3=\delta(Y_3\cup_2 u_2)-\delta Y_3\cup_2 u_2+Y_3\cup_2\delta u_2$ and $\delta Y_3=0$. We can compensate the shift by adding the following coupling to the background $Y_3$

$$-2\pi\frac{p}{2N}\int Y_3\cup_2\delta b_2\ .\tag{B.8}$$

The term is non-trivial since $b_2$ is a $\mathbb{Z}_N$ cochain in general.

To see whether the action is anomalous under gauge transformation, we can extend the fields to the bulk and study how the theory depends on the bulk. A consistent theory requires the bulk term to be independent of the dynamical field $b_2$, and that will give a constraint on the consistent background fields. The total theory depends on the bulk as follows. The action (B.1) contributes the bulk dependence

$$2\pi\frac{p}{2N}\int\left(2b_2\delta b_2+\delta b_2\cup_1\delta b_2\right)\ .\tag{B.9}$$

The additional term (B.8) contributes the bulk dependence

$$-2\pi\frac{p}{2N}\int\left(\delta Y_3\cup_2\delta b_2-Y_3\cup_1\delta b_2-\delta b_2\cup_1 Y_3\right)\ .\tag{B.10}$$

Combining the two contributions and simplify using $\delta b_2-Y_3=0\mod N$ we find the bulk dependence

$$2\pi\frac{p}{N}\int b_2\cup Y_3+2\pi\frac{p}{2N}\int\left(Y_3\cup_1 Y_3-\delta Y_3\cup_2 Y_3\right)\ .\tag{B.11}$$

The first term can be cancelled by demanding $B_2$ to satisfy

$$\delta B_2+pY_3=0\ .\tag{B.12}$$

The remaining bulk dependence together with the contribution from the coupling (B.5) gives the 't Hooft anomaly

$$\frac{2\pi}{N}\int Y_3\cup B_2+2\pi\frac{p}{2N}\int\left(Y_3\cup_1 Y_3-\delta Y_3\cup_2 Y_3\right)\ .\tag{B.13}$$

The anomaly is defined up to local counterterm. For $\gcd(p, N) = 1$ there are integers $\alpha, \beta$ such that $\alpha p = 1 + N\beta$. Then the bulk dependence can be cancelled by the local counterterm

$$\alpha \frac{2\pi}{2N} \int \left( \mathcal{P}(B_2') + pY_3 \cup_2 \delta B_2' \right), \quad B_2' = B_2 + \beta(Np/2)w_2(TM) , \qquad (B.14)$$

where $w_2(TM)$ is the second Stiefel-Whitney class of the tangent bundle. $\delta B_2' = -pY_3$. The local counterterm gives a bulk dependence that cancels the putative 't Hooft anomaly (B.13)

$$\begin{aligned}
&- 2\pi \frac{\alpha p}{N} \int B_2' \cup Y_3 + 2\pi \frac{\alpha p^2}{2N} \int (Y_3 \cup_1 Y_3 - \delta Y_3 \cup_2 Y_3) \\
&= -\frac{2\pi}{N} \int B_2' \cup Y_3 + 2\pi \frac{p(1 + \beta N)}{2N} \int (Y_3 \cup_1 Y_3 - \delta Y_3 \cup_2 Y_3) \\
&= -\frac{2\pi}{N} \int Y_3 \cup B_2 - 2\pi \frac{p}{2N} \int (Y_3 \cup_1 Y_3 - \delta Y_3 \cup_2 Y_3) ,
\end{aligned} \qquad (B.15)$$

where we used $\pi p \beta \int Y_3 \cup_1 Y_3 = \pi p \beta \int w_2 Y_3$, $B_2 Y_3 = Y_3 B_2 + \delta(B_2 \cup_1 Y_3) - \delta B_2 \cup_1 Y_3$, and $\pi p \int (\delta Y_3 / N) \cup_2 Y_3 = -\pi p \int (\delta Y_3 / N) \cup_2 Y_3 \mod 2\pi \mathbb{Z}$.

### B.1.2 Odd $N$

For odd $N$, $p$ is even, and the action (B.1) is independent of the lift of $b_2$ to integral cochain even in the presence of background fields. The action depends on the bulk as

$$\frac{2\pi p}{2N} \int (\delta b_2 b_2 + b_2 \delta b_2) = \frac{2\pi p}{2N} \int (2b_2 Y_3 + Y_3 \cup_1 Y_3) , \qquad (B.16)$$

where we used $\delta b_2 b_2 = b_2 \delta b_2 + \delta(\delta b_2 \cup_1 b_2) + \delta b_2 \cup_1 \delta b_2$ and $\delta b_2 = Y_3$, and we add the $4d$ local counterterm $-(2\pi p/2N) \int \delta b_2 \cup_1 b_2 = -(2\pi p/2N) \int Y_3 \cup_1 b_2$ for nonzero background $Y_3$. A consistent theory requires the dynamical field $b_2$ to be independent of the bulk, and thus the background $B_2$ obeys

$$\delta B_2 + pY_3 = 0 . \qquad (B.17)$$

The 't Hooft anomaly is given by

$$\frac{2\pi}{N} \int Y_3 \cup B_2 + \frac{2\pi p}{2N} \int Y_3 \cup_1 Y_3 . \qquad (B.18)$$

Similarly, for $\gcd(p, N) = 1$ the above bulk dependence can be cancelled by a local counterterm and there is no anomaly.

## B.2 $3d$ $\mathbb{Z}_N$ one-form gauge theory

Consider $\mathbb{Z}_N$ gauge theory with the action

$$\frac{2\pi(k/2)}{N^2} \int b_1 \delta b_1 \ , \tag{B.19}$$

where we will take $k$ to be even and $b_1$ is a $\mathbb{Z}_N$ cocycle.

The theory has one-form symmetries with $\mathbb{Z}_N$ backgrounds $B^e, B^m$. $B^e$ modifies the cocycle condition for $b_1$

$$\delta b_1 = B^e \ , \tag{B.20}$$

while $B^m$ couples as

$$\frac{2\pi}{N} \int b_1 B^m \ . \tag{B.21}$$

In the presence of background $B^e$, since $b_1$ is no longer a $\mathbb{Z}_N$ cocycle, the action (B.19) may not be well-defined. Changing the lift $b_1 \to b_1 + Nu_1$ with integral 1-cochain $u_1$ changes the action by

$$\frac{2\pi(k/2)}{N} \int (u_1 B^e + B^e u_1) \ , \tag{B.22}$$

which can be compensated by adding the following coupling

$$-\frac{2\pi(k/2)}{N^2} \int (b_1 B^e + B^e b_1) \ . \tag{B.23}$$

To examine whether the total action is consistent, we extend the fields to the bulk. The theory is consistent only if the dynamical field $b_1$ is independent of the bulk, and for this to be true the backgrounds are required to obey constraint. The total bulk dependence is

$$2\pi\frac{(k/2)}{N^2} \int (\delta b_1 \delta b_1 - \delta b_1 B^e - B^e \delta b_1 + b_1 \delta B^e - \delta B^e b_1)$$
$$= 2\pi\frac{(k/2)}{N^2} \int (-B^e B^e + 2b_1 \delta B^e + \delta B^e \cup_1 B^e) \ . \tag{B.24}$$

where we used $\delta b_1 = B^e$ mod $N$, $\delta B^e = 0$ mod $N$, and $\delta B^e b_1 = -b_1 \delta B^e - \delta(\delta B^e \cup_1 b_1) - \delta B^e \cup_1 \delta b_1$. The dependence on $b_1$ can be cancelled by demanding $B^m$ to satisfy

$$\delta B^m + k\text{Bock}(B^e) = 0 \ , \tag{B.25}$$

where $\text{Bock}(B^e) = \delta B^e/N$ mod $N$ is the Bockstein homomorphism for the short exact sequence $1 \to \mathbb{Z}_N \to \mathbb{Z}_{N^2} \to \mathbb{Z}_N \to 1$. The backgrounds $B^e, B^m$ with such constraint

describes the one-form symmetry

$$\mathbb{Z}_{\gcd(k,N)} \times \mathbb{Z}_{N^2/\gcd(k,N)} \ . \tag{B.26}$$

The 't Hooft anomaly is given by the remaining bulk dependence, including that contributed from $(2\pi/N) \int b_1 B^m$

$$\frac{2\pi}{N} \int B^e B^m - \frac{2\pi k}{2N^2} \int \mathcal{P}(B^e) \ , \tag{B.27}$$

where $\mathcal{P}(B^e) = B^e \cup B^e - \delta B^e \cup_1 B^e$ is the generalized Pontryagin square of $B^e$.

## B.3  Two-form and one-form coupled gauge theory

Consider

$$2\pi \frac{p}{2N} \int \mathcal{P}(b_2) + \frac{2\pi}{N} \int b_2 \frac{\delta b_1}{N} \ , \tag{B.28}$$

where $b_2, b_1$ are $\mathbb{Z}_N$ cocycles. We turn on backgrounds $Y_2, Y_3, X_2, X_3$, where $X_i$ coupled as

$$\frac{2\pi}{N} \int (b_2 X_2 + b_1 X_3) \ , \tag{B.29}$$

while $Y_2, Y_3$ modifies $b_1, b_2$ to satisfy

$$\delta b_1 = Y_2, \quad \delta b_2 = Y_3 \ . \tag{B.30}$$

In the presence of $Y_2, Y_3$ the second term in (B.28) is no longer well-defined. Consider $b_2 \to b_2 + Nh_2$, $b_1 \to b_1 + Nh_1$ for some integral cochains $h_1, h_2$. This terms shifts by

$$\frac{2\pi}{N} \int (h_2 \delta b_1 + b_2 \delta h_1) = \frac{2\pi}{N} \int (h_2 \delta b_1 - \delta b_2 h_1) = \frac{2\pi}{N} \int (h_2 Y_2 - Y_3 h_1) \ . \tag{B.31}$$

Thus we need to supplement the action with the following coupling to cancel the shift

$$-\frac{2\pi}{N^2} \int (b_2 Y_2 - Y_3 b_1) \ . \tag{B.32}$$

Next we study the bulk dependence of the action coupled to the backgrounds. We find that in order for the dynamical fields $b_1, b_2$ to be independent of the bulk extension, the backgrounds must satisfy

$$\delta X_2 + pY_3 + \text{Bock}(Y_2) = 0, \quad \delta X_3 + \text{Bock}(Y_3) = 0 \ , \tag{B.33}$$

where Bock is the Bockstein homomorphism for $1 \to \mathbb{Z}_N \to \mathbb{Z}_{N^2} \to \mathbb{Z}_N \to 1$.

# C   More general topological field theories

In this appendix, we consider a class of topological field theories that can be defined in any dimension $D$. The degrees of freedom includes a $\mathbb{Z}_N$ $(q+1)$-form gauge field $\widehat{a}$ and a $\mathbb{Z}_N$ $(D-q-1)$-form gauge field $\widehat{b}$. We will use continuous notation that embeds the discrete $\mathbb{Z}_N$ gauge fields in $U(1)$ gauge fields $\widehat{a}$ and $\widehat{b}$. This means that the holonomy $\oint \widehat{a}, \oint \widehat{b} \in \frac{2\pi}{N}\mathbb{Z}$. The action of the theory is

$$S = \int \frac{pN}{2\pi}\widehat{a} \wedge \widehat{b} \, . \tag{C.1}$$

The parameter $p$ has an identification $p \sim p + N$. When $D = 2$, $q = 0$, the theory is equivalent to a $2d$ $\mathbb{Z}_N \times \mathbb{Z}_N$ Dijkgraaf-Witten theory [52].

When $p = 0$, the theory has a $\mathbb{Z}_N$ $(D-q-2)$-form symmetry generated by $\exp(i \oint \widehat{a})$ and a $\mathbb{Z}_N$ $q$-form symmetry generated by $\exp(i \oint \widehat{b})$. We denote their backgrounds by $\widehat{A}_{D-q-1}$ and $\widehat{B}_{q+1}$. The coupling to these backgrounds adds to the action the following term

$$\frac{N}{2\pi} \int (\widehat{a} \wedge \widehat{A}_{D-q-1} + \widehat{B}_{q+1} \wedge \widehat{b}) \, . \tag{C.2}$$

The theory also has a $\mathbb{Z}_N$ $(q+1)$-form symmetry and a $\mathbb{Z}_N$ $(D-q-1)$-form symmetry whose background $\widehat{X}_{q+1}$ and $\widehat{Y}_{D-q}$ modifies the quantization of $\widehat{a}$ and $\widehat{b}$, respectively

$$d\widehat{a} = \widehat{X}_{q+2}, \quad d\widehat{b} = \widehat{Y}_{D-q} \, . \tag{C.3}$$

This implies a mixed anomaly: the coupling to $\widehat{A}_{D-q-1}$ and $\widehat{B}_{q+1}$ is no longer well-defined in the presence of $\widehat{X}_{q+2}$ and $\widehat{Y}_{D-q}$, but depends on the extension to the bulk by

$$\frac{N}{2\pi} \int_{D+1} d\left(\widehat{a} \wedge \widehat{A}_{D-q-1} + \widehat{B}_{q+1} \wedge \widehat{b}\right) = \frac{N}{2\pi} \int_{D+1} \left(\widehat{X}_{q+2}\widehat{A}_{D-q-1} + (-1)^{q+1}\widehat{B}_{q+1}\widehat{Y}_{D-q}\right) \, . \tag{C.4}$$

The anomaly has order $N$ *i.e.* this many copies of the systems has trivial anomaly. To conclude, the theory has a $\mathbb{Z}_N^{(D-q-2)} \times \mathbb{Z}_N^{(q)} \times \mathbb{Z}_N^{(q+1)} \times \mathbb{Z}_N^{(D-q-1)}$ symmetry with a $\mathbb{Z}_N^{(D-q-2)} \times \mathbb{Z}_N^{(q+1)}$ mixed anomaly and a $\mathbb{Z}_N^{(q)} \times \mathbb{Z}_N^{(D-q-1)}$ mixed anomaly. Here $\mathbb{Z}_N^{(q)}$ denotes a $\mathbb{Z}_N$ $q$-form symmetry.

When $p$ is non-trivial, the topological action (C.1) is not well-defined in the prescence of the background $\widehat{X}_{q+2}$ and $\widehat{Y}_{D-q}$. The action has a bulk dependence

$$\frac{pN}{2\pi} \int_{D+1} d(\widehat{a} \wedge \widehat{b}) = \frac{pN}{2\pi} \int_{D+1} \left(\widehat{X}_{q+2} \wedge \widehat{b} + (-1)^{q+1}\widehat{a} \wedge \widehat{Y}_{D-q}\right) \, . \tag{C.5}$$

We can cancel the bulk dependence by modifying the quantization for $\widehat{A}_{D-q-1}$ and $\widehat{B}_{q+1}$ in

the coupling (C.2) to be

$$d\widehat{A}_{D-q-1} + p\widehat{Y}_{D-q} = 0, \quad d\widehat{B}_{q+1} + p\widehat{X}_{q+2} = 0 . \tag{C.6}$$

For $p \neq 1$, this means that $\widehat{X}_{q+2}, \widehat{Y}_{D-q}$ are non-trivial background fields for the higher-form symmetries, but $p\widehat{X}_{q+2}, p\widehat{Y}_{D-q}$ are trivial background gauge fields with holonomy in $2\pi\mathbb{Z}$. Thus the $(q + 1)$-form and the $(D - q - 1)$-form symmetries are broken explicitly to a subgroup by the discrete theta angle.

Another way to see this is that the higher-form symmetry

$$\widehat{a} \to \widehat{a} + \widehat{\lambda}_X, \quad \widehat{b} \to \widehat{b} + \widehat{\lambda}_Y \tag{C.7}$$

changes the topological action (C.1) by

$$\frac{Np}{2\pi} \int \left( \widehat{\lambda}_X b + \widehat{a}\widehat{\lambda}_Y + \widehat{\lambda}_X \widehat{\lambda}_Y \right) \tag{C.8}$$

where $N\widehat{\lambda}_X = q_X d\widehat{\phi}_X$, $N\widehat{\lambda}_Y = q_Y d\widehat{\phi}_Y$ with $q_X, q_Y = 0, \cdots N - 1$. The action is invariant only for $q_X, q_Y \in N\mathbb{Z}/\gcd(N,p)$ and thus the higher-form symmetries are broken to the subgroup $\mathbb{Z}_{\gcd(N,p)}$.

The theory has a putative bulk dependence (C.4). We can reduce it by adding a classical counterterm which shifts the bulk dependence by

$$\frac{Nk}{2\pi} \int_{D+1} d(\widehat{B}_{q+1}\widehat{A}_{D-q-1}) = -\frac{Nkp}{2\pi} \int_{D+1} (\widehat{X}_{q+1}\widehat{A}_{D-q-1} + (-1)^{q+1}\widehat{B}_{q+1}\widehat{Y}_{D-q}) . \tag{C.9}$$

Here $k$ is an arbitrary integer. This reduces the order of the anomaly to $\gcd(N,p)$ *i.e.* this many copies of the systems has trivial anomaly. In particular, when $\gcd(N,p) = 1$, the theory has no anomaly.

To conclude, the theory has an anomaly of order $\gcd(N,p)$, and its backgrounds obey the constraint (C.6).

# D   Gauging $\mathbb{Z}_2 \times \mathbb{Z}_2$ symmetry in $2d$ Ising $\times$ Ising CFT

Orbifold by $\mathbb{Z}_2 \times \mathbb{Z}_2$ symmetry can have discrete torsion since $H^2(\mathbb{Z}_2 \times \mathbb{Z}_2, U(1)) = \mathbb{Z}_2$ [53]. The non-trivial element corresponds to the $\mathbb{Z}_2 \times \mathbb{Z}_2$ Dijkgraaf-Witten theory [52]. Explicitly, denote the two $\mathbb{Z}_2$ gauge fields by $a, a'$ the action is

$$\pi \int a \cup a' . \tag{D.1}$$

The $\mathbb{Z}_2 \times \mathbb{Z}_2$ gauge theory with the Dijkgraaf-Witten action is an invertible bosonic TQFT: the equations of motion for $a, a'$ imply the gauge fields have trivial holonomy.

## D.1  Symmetry in TQFT

Let us study the symmetry of the $\mathbb{Z}_2 \times \mathbb{Z}_2$ Dijkgraaf-Witten theory. The backgrounds $B_2, B_2'$ for the one-form symmetry modify the fluxes of the gauge fields

$$\delta a = B_2, \quad \delta a' = B_2' . \tag{D.2}$$

Let $B_1, B_1'$ denote the backgrounds for the 0-form symmetries generated by $\oint a, \oint a'$. They couple to the theory through

$$\pi \int a \cup B_1 + \pi \int B_1' \cup a' . \tag{D.3}$$

The coupling $\pi \int a \cup a'$ are not well-defined in the presence of $B_2, B_2'$, but it can be cancelled by an analogue of the Green-Schwarz mechanism

$$\delta B_1 = B_2', \quad \delta B_1' = B_2 . \tag{D.4}$$

The coupling (D.3) has a bulk dependence for the background fields, but it can be cancelled by the local counterterm of backgrounds $\pi \int B_1 \cup B_1'$:

$$\pi \int \delta a \cup B_1 + B_1' \cup \delta a' = \pi \int \delta(B_1' \cup B_1) . \tag{D.5}$$

## D.2  Coupling CFT to TQFT

An an example, consider Ising $\times$ Ising conformal field theory (CFT) in $(1 + 1)d$. The theory has a $\mathbb{D}_8$ 0-form symmetry that includes a $\mathbb{Z}_2 \times \mathbb{Z}_2$ non-anomalous subgroup. In the following, we will discuss gauging the symmetry with or without discrete torsion *i.e.* a $\mathbb{Z}_2 \times \mathbb{Z}_2$ Dijkgraaf-Witten theory.

An Ising CFT has three Virasoro primaries including the vacuum operator 1 with $h = \overline{h} = 0$, the energy operator $\epsilon$ with $h = \overline{h} = \frac{1}{2}$ and a spin field $\sigma$ with $h = \overline{h} = \frac{1}{16}$. The theory has a $\mathbb{Z}_2$ symmetry that flips the spin fields:

$$\mathbb{Z}_2: \quad 1 \to 1, \quad \epsilon \to \epsilon, \quad \sigma \to -\sigma . \tag{D.6}$$

The torus partition function of the Ising model is the sum of the characters of the three

parimaries

$$Z_{\text{Ising}}(\tau, \overline{\tau}) = |\chi_0(\tau)|^2 + |\chi_{\frac{1}{2}}(\tau)|^2 + |\chi_{\frac{1}{16}(\tau)}|^2 \ . \tag{D.7}$$

The characters are

$$\chi_0(\tau) = \frac{1}{2}\left(\sqrt{\frac{\theta_3(\tau)}{\eta(\tau)}} + \sqrt{\frac{\theta_4(\tau)}{\eta(\tau)}}\right), \quad \chi_{\frac{1}{2}}(\tau) = \frac{1}{2}\left(\sqrt{\frac{\theta_3(\tau)}{\eta(\tau)}} - \sqrt{\frac{\theta_4(\tau)}{\eta(\tau)}}\right), \quad \chi_{\frac{1}{16}}(\tau) = \sqrt{\frac{\theta_2(\tau)}{2\eta(\tau)}}, \tag{D.8}$$

where the $\theta_i$ are the Jacobi theta function, defined as

$$\theta_2(\tau) = 2\sum_{n=1}^{\infty} q^{\frac{1}{2}(n-\frac{1}{2})^2} \ ,$$

$$\theta_3(\tau) = 1 + 2\sum_{n=1}^{\infty} q^{n^2/2} \ , \tag{D.9}$$

$$\theta_4(\tau) = 1 + 2\sum_{n=1}^{\infty} (-1)^n q^{n^2/2} \ ,$$

and $\eta$ is the Dedekind eta function defined as

$$\eta(\tau) = q^{1/24}\prod_{n=1}^{\infty}(1 - q^n) \ . \tag{D.10}$$

Here $q = e^{2\pi i \tau}$. Inserting the $\mathbb{Z}_2$ symmetry lines along the temporal and the spatial directions leads to another three torus partition functions

$$Z_{\text{Ising}}^{H}(\tau, \overline{\tau}) = |\chi_0(\tau)|^2 + |\chi_{\frac{1}{2}}(\tau)|^2 - |\chi_{\frac{1}{16}(\tau)}|^2$$

$$Z_{\text{Ising}}^{V}(\tau, \overline{\tau}) = \chi_0(\tau)\chi_{\frac{1}{2}}(\overline{\tau}) + \chi_{\frac{1}{2}}(\tau)\chi_0(\tau) + |\chi_{\frac{1}{16}(\tau)}|^2 \tag{D.11}$$

$$Z_{\text{Ising}}^{HV}(\tau, \overline{\tau}) = -\chi_0(\tau)\chi_{\frac{1}{2}}(\overline{\tau}) - \chi_{\frac{1}{2}}(\tau)\chi_0(\tau) + |\chi_{\frac{1}{16}(\tau)}|^2$$

Let us gauge the $\mathbb{Z}_2$ symmetry of the Ising CFT. The torus partition function of the orbifold theory

$$Z_{\text{gauged Ising}}(\tau, \overline{\tau}) = \frac{1}{2}\left(Z_{\text{Ising}} + Z_{\text{Ising}}^{H} + Z_{\text{Ising}}^{V} + Z_{\text{Ising}}^{HV}\right) = Z_{\text{Ising}}(\tau, \overline{\tau}) \ , \tag{D.12}$$

is the same as the torus partition function of an Ising CFT. This implies that the $\mathbb{Z}_2$ orbifold of an Ising CFT is again an Ising CFT. The orbifold theory has a $\mathbb{Z}_2$ symmetry which can be viewed as the emergence dual $\mathbb{Z}_2$ symmetry of the gauged $\mathbb{Z}_2$ symmetry.

Now consider two copies of Ising CFTs. The theory has a $\mathbb{Z}_2 \times \mathbb{Z}_2$ symmetry that flips the spin fields $\sigma$ in one of the two copies. It also has a $\mathbb{Z}_2$ symmetry that swaps the two

copies. These two symmetries combine into a $\mathbb{D}_8$ symmetry.

The $\mathbb{Z}_2 \times \mathbb{Z}_2$ orbifold of the theory is also an Ising×Ising CFT. The orbifold theory has a $\mathbb{D}_8$ symmetry whose $\mathbb{Z}_2 \times \mathbb{Z}_2$ subgroup are the emergent dual symmetry of the gauged $\mathbb{Z}_2 \times \mathbb{Z}_2$ symmetry while the $\mathbb{Z}_2$ symmetry that exchanges the two copies remains intact.

The $\mathbb{Z}_2 \times \mathbb{Z}_2$ orbifold theory can include a discrete torsion. This modifies the torus partition function into

$$
\begin{aligned}
Z^{\text{torsion}}_{\text{gauged Ising}^2}(\tau, \overline{\tau}) = {} & \frac{1}{4} Z_{\text{Ising}} \left( Z_{\text{Ising}} + Z^H_{\text{Ising}} + Z^V_{\text{Ising}} + Z^{HV}_{\text{Ising}} \right) \\
& + \frac{1}{4} Z^H_{\text{Ising}} \left( Z_{\text{Ising}} + Z^H_{\text{Ising}} - Z^V_{\text{Ising}} - Z^{HV}_{\text{Ising}} \right) \\
& + \frac{1}{4} Z^V_{\text{Ising}} \left( Z_{\text{Ising}} - Z^H_{\text{Ising}} + Z^V_{\text{Ising}} - Z^{HV}_{\text{Ising}} \right) \\
& + \frac{1}{4} Z^{HV}_{\text{Ising}} \left( Z_{\text{Ising}} - Z^H_{\text{Ising}} - Z^V_{\text{Ising}} + Z^{HV}_{\text{Ising}} \right) \\
= {} & Z^{r=1}_{\text{compact boson}}(\tau, \overline{\tau}) \,,
\end{aligned}
\tag{D.13}
$$

which is the same as the torus partition function of a compact boson with radius $r = 1$ $i.e.$ $U(1)_4$. We choose the convention that the self-dual radius is $r = 1/\sqrt{2}$.

To summarize,

- Ising $\times$ Ising gauging $\mathbb{Z}_2 \times \mathbb{Z}_2$ without discrete torsion: Ising $\times$ Ising.

- Ising $\times$ Ising gauging $\mathbb{Z}_2 \times \mathbb{Z}_2$ with discrete torsion: compact boson $U(1)_4$.

We remark that orbifold with discrete torsion can also be understood as a two step gauging process. First we gauge the $Z_2$ symmetries in the first Ising CFT. Second we gauge the diagonal $Z_2$ symmetry of the second Ising CFT and the orbifold theory of the first Ising CFT.

$$
Z^{\text{torsion}}_{\text{gauged Ising}^2} = \sum_{a,b} Z_{\text{Ising}}[a] Z_{\text{Ising}}[b] \exp\left( i\pi \int a \cup b \right) = \sum_{b} Z_{\text{Ising}}[b] Z_{\text{gauged Ising}}[b] \tag{D.14}
$$

Since the orbifold of an Ising CFT is itself, this amounts to gauging the diagonal $\mathbb{Z}_2$ symmetry of two copies of Ising CFTs. The resulting theory is $U(1)_4$ [54].

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
