# Peer review of "Discrete Theta Angles, Symmetries and Anomalies"

_SciPost Physics_

## Round 2 · Referee Report · Anonymous (Referee 1) · 2020-10-7

Strengths

The article contains new and interesting results, especially on two-group symmetries and mixed anomalies.

Weaknesses

  1. The style of writing is rather terse.
  2. The article was written such that the target readers are experts in the field of anomalies. Certain concepts, such as the symmetry fractionalization, in this article could be better explained or reviewed.

Report

This article studied gauge theories with different discrete theta angles by gauging a global symmetry with different symmetry-protected topological (SPT) phases. The authors observed that when the symmetries in question are of higher-form with different degrees, gauging a subgroup symmetry that does not have a mixed anomaly with the remaining symmetry could lead to theories with two-group symmetries. As a result, two-group symmetries studied in this article, in general, depend on the discrete theta angles of the gauge theory. The authors also studied various mixed anomalies. For example, one of the interesting results is in section 4.5 where it was shown that, in the SO(N) QCD, there is a mixed anomaly between the magnetic $\mathbb{Z}_2$ one-form symmetry, the flavor symmetry and the charge conjugation symmetry. In the referee's opinion, the article deserves publication after minor improvements on the presentation.

Requested changes

  1. The authors should explain or motivate in physical terms the following concepts: the Bockenstein homomorphism and the symmetry fractionalization. This would make the article more accessible to a wider audience.
  2. In the article, the authors omitted the discussion on the groups $Sp(N)$ and $Sp(N)/\mathbb{Z}_2$. For the completeness of the paper, the authors should mention how their techniques can be applied to such cases.

  • validity: high
  • significance: high
  • originality: high
  • clarity: good
  • formatting: perfect
  • grammar: perfect

Author:  Ho Tat Lam  on 2020-11-06  [id 1035]

(in reply to Report 1 on 2020-10-07)

We thank the referee for comments and suggestions! Please see the reply below.

  1. We will add paragraphs elaborating on the two concepts and point the readers to relevant references.

  2. Unlike the $SU(N)$, $Spin(N)$ gauge group discussed in the paper, $Sp(N)$ gauge group only has a $\mathbb{Z}_2$ center which does not have any nontrivial proper subgroup. Hence $Sp(N)$ or $Sp(N)/\mathbb{Z}_2$ gauge theories with matters do not have two-group symmetries with nontrivial Postnikov class. Therefore, we didn't discuss them in the paper. We will add a footnote explaining this.

---

## Round 2 · Referee Report · Anonymous (Referee 2) · 2021-1-22

Report

This is an interesting paper in the context of the study of generalized notions of symmetries and anomalies. The main focus of this paper are the general relation among families of gauge theories with different discrete theta angle, their global symmetries and ’t Hooft anomalies.

Theories with different discrete theta angles often arise from gauging a global symmetry in a quantum field theory with different symmetry-protected topological (SPT) phases. The main point of this paper is arguing how two inequivalent choices of gauging are related via coupling to a TQFT, that the authors explicitly identify.

Several examples of this including gauge theories with or without matter in 3d and 4d are presented in this paper, as well as consistency checks with other results in the relevant literature, that can be reproduced from this more advanced perspective.

The authors assume their readers have a good knowledge of the relevant mathematics, which might make this paper a hard read for a neophyte, but this is understandable, as the subject is in rapid development, and so are the required technical tools.

We find this paper is interesting, well-written and original. It is a good fit for the journal, and we strongly recommend accepting it for publication.

---

## Editorial Decision

resubmitted